# Location- and feature-based selection histories make independent, qualitatively distinct contributions to urgent visuomotor performance

Emily E Oor[1], Emilio Salinas[2]*, Terrence R Stanford[2]*

[1]Department of Psychology, Wake Forest University, Winston-Salem, United States; [2]Department of Translational Neuroscience, Wake Forest University School of Medicine, Winston Salem, United States

## eLife Assessment

Oor and colleagues report the potentially independent effects of the spatial and feature-based selection history on visuomotor choices. They outline **compelling** evidence, tracking the dynamic history effects based on their extremely clever experimental design (urgent version of the search task). Their finding is of **fundamental** significance, broadening the framework to identify variables contributing to choice behavior and their neural correlates in future studies.

**\*For correspondence:**
esalinas@wakehealth.edu (ES);
stanford@wakehealth.edu (TRS)

**Abstract** Attention mechanisms guide visuomotor behavior by weighing physical salience and internal goals to prioritize stimuli as choices for action. Although less well studied, selection history, which reflects multiple facets of experience with recent events, is increasingly recognized as a distinct source of attentional bias. To examine how selection history impacts saccadic choices, we trained two macaque monkeys to perform an urgent version of an oddball search task in which a red target appeared among three green distracters or vice versa. By imposing urgency, performance could be tracked continuously as it transitioned from uninformed guesses to informed choices as a function of processing time. This, in turn, permitted assessment of attentional control as manifest in motor biases, processing speed, and asymptotic accuracy. Here, we found that the probability of making a correct choice was strongly modulated by the histories of preceding target locations and target colors. Crucially, although both effects were gated by success (or reward), their dynamics were clearly distinct: whereas location history promoted a motor bias, color history modulated perceptual sensitivity, and these influences acted independently. Thus, combined selection histories can give rise to enormous swings in visuomotor performance even in simple tasks with highly discriminable stimuli.

## Introduction

The choice of where to look, which is integral to most daily human activities, reflects dynamic interplay between information sensed from the external environment and internal goals specific to the task at hand. This interaction is likely a key contributor to the variability in choice and associated outcome that typifies real-world behavior. But even in a laboratory setting, performance on visuomotor choice tasks varies not only between subjects but also for the same subject across nominally identical sessions and trials. Understanding the drivers of behavioral variance is essential to determining the factors that limit performance and, ultimately, to deducing their neural basis.

It is a given that attentional state contributes substantially to the conditions that determine success or failure on any given goal-oriented task. Endogenous mechanisms allocate attention volitionally, prioritizing choices according to internal goals, whereas exogenous mechanisms attract attention automatically in proportion to the physical saliencies of external features or events (*Itti and Koch, 2001*; *Theeuwes, 2010*; *Carrasco, 2011*; *Wolfe and Horowitz, 2017*). How strongly these mechanisms engage their respective neural substrates and the degree to which these attentional pointers align or conflict contribute to determining if, when, and how any particular choice is made (*Busse et al., 2008*; *Markowitz et al., 2011*; *Chen et al., 2013*; *Scerra et al., 2019*; *Chen et al., 2020*; *Goldstein et al., 2022*; *Seideman et al., 2022*; *Oor et al., 2023*).

Although exogenous and endogenous attentional mechanisms are potent influences, physical stimulus properties and explicit knowledge of task requirements do not solely determine choice performance in any given instance. It is now generally accepted that current choices are constrained by the recent history of past choices in ways that are neither consistent with task goals nor a straightforward reflection of the physical saliencies of available options (*Fecteau and Munoz, 2003*; *Awh et al., 2012*; *Theeuwes, 2018*; *Theeuwes, 2019*; *Anderson et al., 2021*). 'Selection history' is the umbrella term for attention mechanisms that are distinct from stimulus-driven and goal-directed forms in that they comprise a collection of implicit biases shaped by past experience. Potential sources of selection history bias include feature-based priming (*Maljkovic and Nakayama, 1994*; *McPeek et al., 1999*; *Kristjánsson and Campana, 2010*; *Brascamp et al., 2011*; *Kristjánsson and Ásgeirsson, 2019*), location bias (*Maljkovic and Nakayama, 1996*; *Chelazzi et al., 2014*; *Hickey et al., 2014*), and stimulus- or action-reward associations (*Barraclough et al., 2004*; *Hikosaka et al., 2006*; *Anderson et al., 2011*; *Abrahamyan et al., 2016*; *Hauser et al., 2018*; *Anderson, 2019*; *Hermoso-Mendizabal et al., 2020*; *Lak et al., 2020*), among others. Given their heterogeneity, selection history biases are likely to impact the coding of both intrinsic and extrinsic decision variables – and do so on multiple timescales – to drive choice trends (*Awh et al., 2012*; *Theeuwes, 2018*; *Theeuwes, 2019*; *Anderson et al., 2021*).

Although selection history influences are measurable as effects on performance accuracy, reaction time (RT), or both, discerning how multiple sources of variance contribute to determining choice outcome is not straightforward. Just as these implicit biases need not align with goal-directed or stimulus-based drivers of attention, different sources of selection history bias may reinforce or conflict with each other to varying degrees according to their respective history-dependent profiles. Thus, via internal trade-offs, many different combinations of selection history bias could yield very similar behavioral outcomes as measured by accuracy and RT. Moreover, because history-induced biases are not explicitly strategic – they develop even when history offers no predictive value to the current trial – their expression in measures of overt behavior will be attenuated in laboratory tasks that prioritize accuracy and allow sufficient time for choices to be guided predominantly by goal-driven mechanisms.

The current study uses urgency (*Stanford and Salinas, 2021*) to isolate and quantify how selection history biases deriving from target feature (color) and target location impact saccadic choices; specifically, the ability to identify and look toward a color singleton (i.e., an 'oddball' stimulus of unique color) presented within an array of uniform distracters on each trial. With the so-called compelled oddball (CO) task (*Scerra et al., 2019*), the goal was to differentiate sources of history bias based on both inter- and intra-trial temporal dynamics and, having done so, determine the degree to which they act independently or interact in biasing each choice. Urgency is the key, first, for maximizing the measurable impact of bias, and second, for distinguishing the influences of biases that act on stimulus features (e.g., color priming) from those that do not (e.g., location priming). This is possible because, under time pressure, individual choices can be precisely identified via their unique temporal signatures as either uninformed guesses or perceptually informed decisions (*Salinas and Stanford, 2021*; *Stanford and Salinas, 2021*), and spatial and perceptual biases are predicted to manifest in very different ways for each. Motor biases based on target location history are expected to be most evident either before or soon after presentation of the stimulus array (which is when guesses are elicited), and to wane thereafter, once perceptual information becomes available. In contrast, biases based on target color history should be completely absent early on and become evident only after some processing of the stimulus feature content has occurred (which is when informed decisions are elicited).

Here, we identify these distinct sources of bias in the performance of rhesus monkey subjects, thus laying the groundwork for future neurophysiological studies aiming to reveal their neural basis. Our

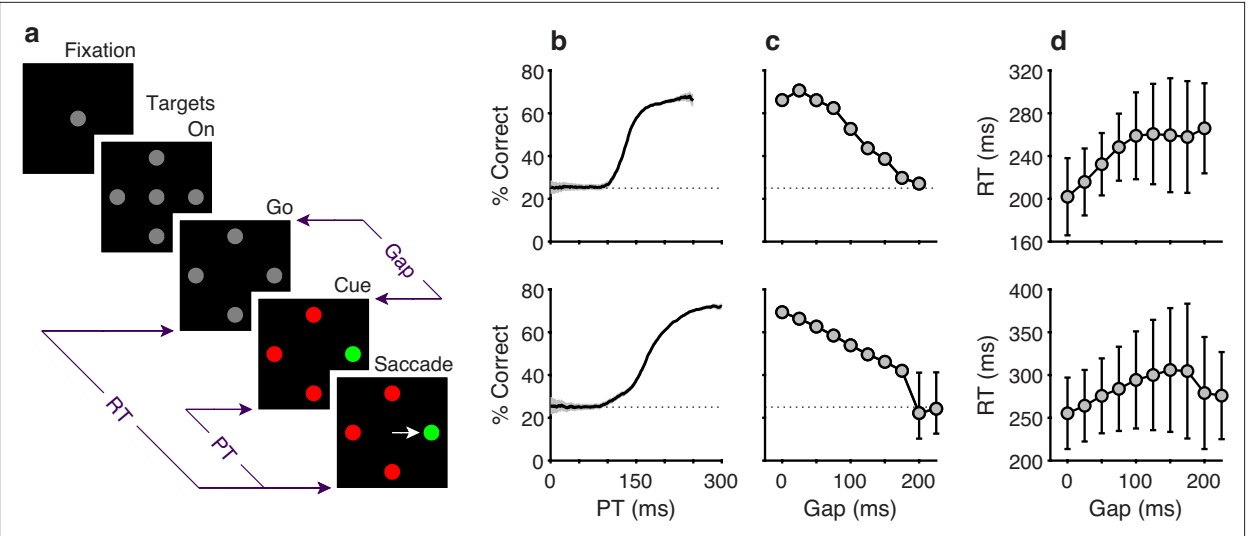

**Figure 1.** Behavioral task and psychophysical performance from two monkeys. (**a**) The compelled oddball (CO) task. Trials begin when the subject fixates on the central gray spot (Fixation). Four gray placeholders then appear (Targets On) separated by 90° surrounding the fixation spot (at variable eccentricities and orientations across days). Urgency is incorporated by next extinguishing the fixation spot (Go), which instructs the subject to respond. After a variable time delay (Gap, 0–225 ms), the target and distracter identities are revealed (Cue). A correct choice requires a saccade to the stimulus with a unique color, the oddball stimulus, within 450 ms of the go signal. The reaction time (RT) is measured from the go signal to saccade onset. The processing time (PT) interval is the maximum amount of time during which the cue can inform the choice. (**b**) Tachometric curves showing the percentage of correct responses as a function of PT for monkey C (top; 26,038 trials) and monkey N (bottom; 45,173 trials). Gray shades represent 95% confidence intervals (CIs). Chance performance (25%) is indicated by dotted lines. PT bin size is 50 ms. (**c**) Mean percentage of correct choices as a function of gap for monkey C (top) and monkey N (bottom). Error bars indicate 95% CIs. (**d**) Mean RT as a function of gap for monkey C (top) and monkey N (bottom). Error bars indicate ±1 standard deviation.

The online version of this article includes the following figure supplement(s) for figure 1:

**Figure supplement 1.** Variability in performance across animals and rule sets.

findings demonstrate that spatial and feature-based selection histories exert independent influences on behavior, and that their combined impact can be extraordinarily large even for visuomotor tasks that involve simple rules and highly discriminable stimuli.

## Results

### Accuracy of urgent singleton search depends on perceptual processing time

Two monkeys performed the CO task (*Figure 1a*), a singleton search paradigm that, via the imposition of a response time limit, compels subjects to begin preparing a response in advance of making the perceptual judgment that determines the correct choice (Methods). In this task, urgency is incorporated by delivering the signal to make a saccade (*Figure 1a*, Go) before presenting the target-distracter stimulus array, or color cue (*Figure 1a*, Cue); this cue is withheld for a variable amount of time (*Figure 1a*, Gap). By design, the task mandates that saccades be initiated within a normal RT range (RT ≤ 450 ms). This, together with the variable Gap interval, yields choices based on a wide range of nominal cue viewing times or processing times (PTs, where PT is calculated by subtracting RT – Gap in each trial).

In practice, both monkeys performed the task as designed, demonstrating behavioral trends qualitatively consistent with those described and modeled in previous studies that used two-alternative urgent tasks (*Stanford et al., 2010*; *Salinas et al., 2010*; *Shankar et al., 2011*). Plotted for each Gap, average RTs ranged between 200 and 250 ms for monkey C (*Figure 1d*, top) and 250 and 300 ms for monkey N (*Figure 1d*, bottom), while mean choice accuracy values changed gradually (*Figure 1c*). Importantly, the tachometric curve, which plots performance as a function of PT (*Figure 1b*), shows that choice accuracy rises sharply as the amount of cue viewing time increases. Thus, PT readily

distinguishes between choices that are likely to be guided by the color cue (informed) and those that are not (guesses). For PT < 100 ms, the performance of both monkey subjects was at chance (probability correct = 0.25), indicating that such choices were not informed by the color cue. Conversely, for PT ≳ 150 ms for monkey C, and PT ≳ 200 ms for monkey N, most choices were informed, as indicated by performance asymptotes that approached 70% correct. We refer to the performance levels at each of these PT extremes as the floor and ceiling accuracies, respectively (Methods).

The measured ceiling levels seem unusually low for a visuomotor task in which the response rule is so simple and the stimuli so highly discriminable. Why? Two observations already point to selection histories as responsible for such low numbers. First, the results were unlikely due to insufficient training. This is because, although our subjects practiced the CO task extensively (*Figure 1b*, caption) throughout several months of data collection, their ceiling accuracies did not improve (the Pearson correlation between experimental session and ceiling performance was 0.2, p=0.1 for monkey C and 0.05, p=0.6 for monkey N). And second, urgency itself is also unlikely to be a significant factor limiting the accuracy of the monkeys' informed choices. This is because other animals from our laboratory reached high ceilings (~95% correct) in the urgent CO task when the target color remained fixed or when it switched only between blocks of trials (*Figure 1—figure supplement 1*; *Scerra et al., 2019*). Together, these two observations suggest that what makes the CO task particularly difficult is the randomization of the target colors and locations. Indeed, the same conclusion would be reached based on prior studies that used standard nonurgent versions of the oddball task; in that case, monkey performance was also much lower when both color and location were randomized (*Song and McPeek, 2015*) than when the target colors alternated between blocks of trials (*Bichot and Schall, 1999*; *Bichot and Schall, 2002*) or between days (*McPeek and Keller, 2004*).

As will be shown next, when both the target color and location change unpredictably across trials, performance during oddball search is swayed by selection histories to a remarkable degree.

## Perceptual and non-perceptual selection history biases are dissociable on the basis of PT

The ability to differentiate informed from uninformed choices is the basis for distinguishing the influences of selection history biases that derive from perceptual versus non-perceptual sources. If present, perceptual priming due to target color repetition could, in principle, affect the timing, slope, and/or ceiling of the tachometric curve, each an attribute that reflects the likelihood that a choice is perceptually informed (*Shankar et al., 2011*). Perceptual priming cannot, however, influence uninformed choices – saccades made at short PTs – before the processing of color information has had any impact. Importantly, these perceptually uninformed choices are precisely those predicted to be most sensitive to non-perceptual biases. In the context of an urgent feature discrimination task, motor plans developing in advance of cue information (i.e., during the Gap interval) might be expected to incorporate spatial biases that reflect the recent history of target locations. If so, the expression of a history-induced location bias would be maximal for guesses and diminish as choices become increasingly guided by perceptual evidence.

Because the effects of selection history on performance were highly consistent for the two monkeys, in this and the following subsections, these effects are presented and discussed based on the data pooled across subjects. We return to the differences and similarities between the two individuals in more detail in the last section of the Results, once the main phenomena have been characterized.

The impacts of target color (*Figure 2a–d*) and target location (*Figure 2e–h*) repetition on performance were assessed by post hoc sorting of trials based on their respective histories (Methods). Separate tachometric curves were computed from trials conditioned on the number of preceding trials (1–4) for which either target color (*Figure 2a*) or target location (*Figure 2e*) was the same (S) or different (D). For example, the 1S condition includes the color sequences red-**red** and green-**green** (bold type indicates the target color on the trial that contributes to the given tachometric curve), while the 3S condition includes the color sequences red-red-red-**red** and green-green-green-**green**. Likewise, D trials are preceded by *N* trials with a target of the opposing color. The 1D condition, then, includes the color sequences green-**red** and red-**green**, while the 3D condition includes the color sequences green-green-green-**red** and red-red-red-**green**. The same nomenclature was applied to target location repetitions (Methods). In this case, S trials are preceded by *N* (1–4) trials having the same target location (e.g., right-right-**right** or down-down-**down**), and D trials are those preceded by

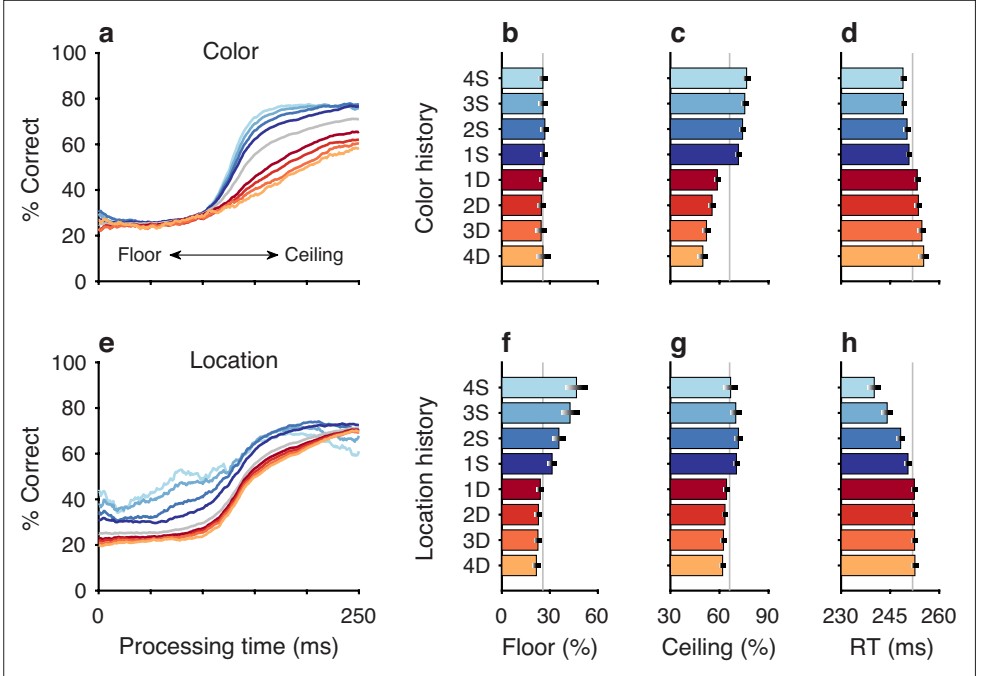

**Figure 2.** Repetitions and switches of target color and target location cause distinct modulations of performance in the compelled oddball (CO) task. Each panel shows results for performance on the current trial conditioned on the history of preceding trials when current and past trials are the same (S, blue spectrum; color or location repeats) or different (D, red spectrum; color or location switches). Both S and D sequences go back 1–4 trials into the past. (**a**) Tachometric curves conditioned on target color history. Performance is shown as a function of processing time (PT) for eight different history sequences. Average performance is shown in gray. Bin width is 50 ms for all curves. For clarity, error bars are omitted. (**b**) Choice accuracy (x axis, percent correct) in short-PT trials (PT <100 ms, floor) conditioned on color history (y axis). Error bars represent 95% confidence intervals (CIs) across trials. Gray vertical line denotes value for the average tachometric curve. (**c**) Choice accuracy in long-PT trials (PT >150 ms, ceiling) conditioned on color history. Same format as in b. (**d**) Average reaction time (RT) (x axis, in ms) conditioned on color history (y axis). Error bars show ±1 standard error of the mean (SEM). The gray vertical line denotes average RT across all trials. (**e**) Tachometric curves conditioned on target location history. (**f**) Choice accuracy in short-PT trials (PT <100 ms, floor) conditioned on location history. (**g**) Choice accuracy in long-PT trials (PT >150 ms, ceiling) conditioned on location history. (**h**) Average RT conditioned on location history.

The online version of this article includes the following figure supplement(s) for figure 2:

**Figure supplement 1.** Target color repetition affects perceptual accuracy in a consistent manner across subjects.

**Figure supplement 2.** Target color repetition affects decision accuracy in a consistent manner across colors.

---

N trials in which the singleton target appeared at any one of the three alternate locations (e.g., left-down-**right** or up-up-**left**).

After combining the data from the two monkey subjects (Methods; *Figure 2—figure supplement 1*), tachometric curves were generated conditioned on the specific histories just described (*Figure 2a and e*). Strong selection history effects became evident, as performance depended systematically on the number of preceding trials in which the target color (*Figure 2a–d*) or the target location (*Figure 2e–i*) was either the same or different from that in the current trial. But critically, the respective influences of these two history variables on choice behavior were readily distinguishable by their PT dependencies, as anticipated.

The influence of target color repetition only became apparent at PT > 100 ms, an inflection point that marks the transition from uninformed choices to those increasingly steered by color information. From there, each additional preceding trial of the same target color (1S–4S) was associated with a performance increment, as evident in both the slopes and ceiling values of the corresponding tachometric curves (*Figure 2a*, blue curves; *Figure 2c*, blue bars). Correspondingly, decreasing slopes and ceiling values depending on the number of preceding trials of a different target color (*Figure 2a*, red curves; *Figure 2c*, red bars) were indicative of history-induced performance decrements. Thus, the

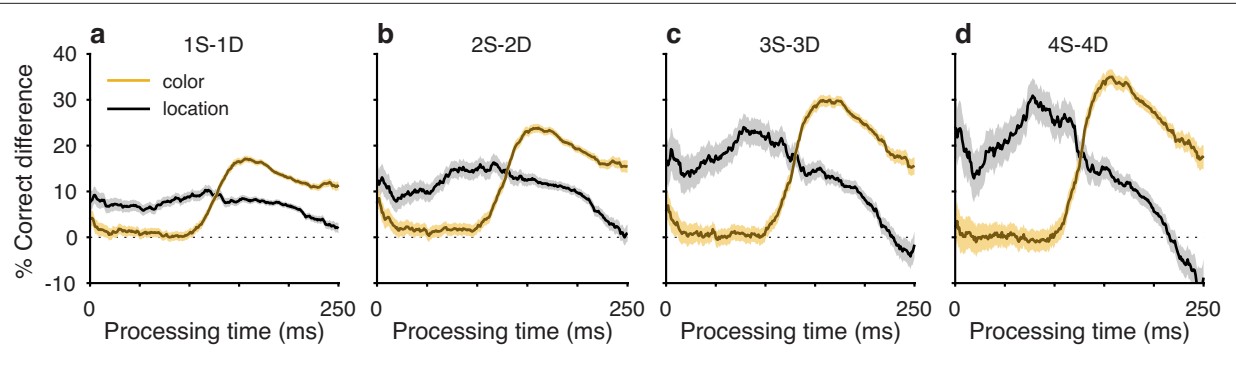

**Figure 3.** History effects due to target color and target location have distinct temporal manifestations within the time course of a trial. Each curve plots the difference between two tachometric curves, one conditioned on same (S) trials and another conditioned on different (D) trials. Each panel shows two curves, one for color history (gold traces) and another for location history (black traces), with shaded error bands indicating ± 1 SE across trials. Dotted lines at zero mark no difference in performance given S and D histories. (**a**) Difference curves for histories going back 1 trial (1S – 1D). (**b**) Difference curves for histories going back 2 trials (2S – 2D). (**c**) Difference curves for histories going back 3 trials (3S – 3D). (**d**) Difference curves for histories going back 4 trials (4S – 4D). Note that the location bias is strongest at short processing times (PTs) and then declines, whereas the color bias rises sharply after ~100 ms of PT.

trial-by-trial history of target color decreased (same color) or increased (different color) the amount of cue viewing time needed to achieve a given level of accuracy. Importantly, this perceptual bias altered the efficacy with which color information guided the choice, but, as expected, had no measurable effect on uninformed choices, as indicated by uniformity of the floor accuracies across the range of history conditions (*Figure 2a and b*).

In contrast, the principal effect of target location repetition was on uninformed choices (*Figure 2e and f*), those made at PT < 100 ms, before cue information began to guide behavior. For these perceptually uninformed choices, or guesses, the likelihood of looking toward a given location increased as a function of the number of preceding trials in which the target appeared at that location (*Figure 2e*, blue curves; *Figure 2f*, blue bars) and decreased slightly when preceding targets appeared elsewhere (*Figure 2e*, red curves; *Figure 2f*, red bars).

The PT-based dichotomy between color and location history biases and their respective selection history dependencies (i.e., effect of number of S or D trials) is well summarized by a contrast computed as the difference between two tachometric curves, one conditioned on a given number of S repetitions minus another conditioned on the same number of D repetitions (*Figure 3*). Thus, for color (gold traces), the resulting difference curve reveals a PT-dependent bias that tracks closely with the increasing likelihood of making a correct color discrimination; the bias is near zero for uninformed choices, but its magnitude rises sharply after the inflection point (at PT $\gtrsim$ 100 ms) that demarcates the transition to increasingly informed choices. In contrast, for location (black traces), the corresponding bias demonstrates the inverse trend: it is strongest for uninformed choices and decreases progressively after the inflection point. For both color and location bias, the size of the computed differential trend increases monotonically as the number of S and D repetitions increases from one (*Figure 3a*, 1S–1D) to four (*Figure 3d*, 4S–4D).

An important consideration here is that these biases cannot be explained by differences in sensitivity (or preference) between red and green targets, because history effects were qualitatively similar for targets of each separate color (*Figure 2—figure supplement 2*).

In summary, the results so far validate the key qualitative prediction – that selection history biases modulate performance in the CO task substantially, with perceptual and non-perceptual components being dissociable based on the characteristic ways in which they impact the temporal evolution of the choice process. These two components were characterized further.

## Color- and location-driven biases have different inter-trial timescales

Biases accumulate across trials (*Figure 3*), indicating that the influence of a unitary color or location event persists over time (or trials). Notably, the duration of this influence, or analogously, the time course over which it wanes, is another potential criterion for distinguishing the biases induced by

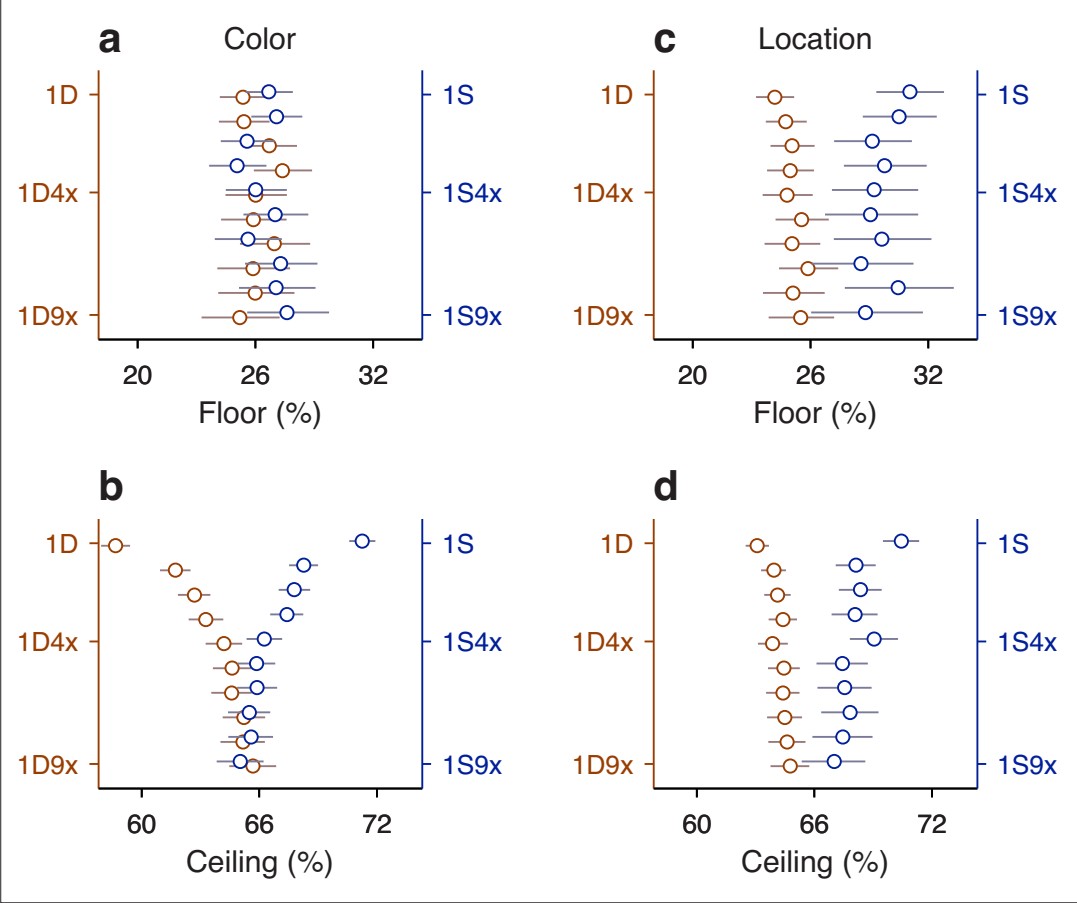

**Figure 4.** The timescale of history effects induced by a single repeat or a single switch varies across history types. (**a**) Task performance during uninformed choices (floor, percent correct) following a single-color repetition (1S; blue axis) or color switch (1D; brown axis) that occurred 1–10 trials prior to the trial being evaluated. The notation Nx indicates N intervening trials of any type between the first event and the repeat or switch. (**b**) As in a, but for performance during informed choices (ceiling). (**c**) Task performance during uninformed choices (floor, percent correct) following a target location repetition (1S, blue axis) or target location switch (1D, red axis) that occurred 1–10 trials prior to the trial being evaluated. (**d**) As in c, but for performance during informed choices (ceiling). In all plots, error bars represent 95% confidence intervals (CIs).

color and location selection histories. We quantified the impact on performance of a single repeat (in S sequences) or switch (in D sequences) for both color (*Figure 4a and b*) and location (*Figure 4c and d*) by computing tachometric curves from trials that occurred N=1 to N=10 trials after a reference event (Methods). For example, for color, the condition 1S4x denotes a color repeat separated by 5 trials and includes two possible sequences, red-xxxx-**red** and green-xxxx-**green**, where the color of the first trial is the same as that of the trial being evaluated (bolded) and each of the four intervening trials can be of either color (x). Similarly, for location, the condition 1D2x denotes a location switch separated by 2 intervening trials of any type and would include sequences such as up-xx-**down** or left-xx-**right**, for instance.

Note that, although the 1S and 1D conditions are identical for this and the preceding analyses, the resulting trends as functions of sequence length should be opposite. When evaluating the effect of increasing repetitions (*Figure 2*), the difference between, say, 4S and 4D conditions is generally larger than that between 1S and 1D. In contrast, when evaluating the effect of a single event (*Figure 4*), the difference between 1S4x and 1D4x conditions should be generally smaller than that between 1S and 1D, as the reference event becomes more distant from the current choice.

As anticipated, a single-color event, whether corresponding to a repeat or a switch, had no impact on a future uninformed choice, as indicated by the floors of the tachometric curves obtained for different trial separations (*Figure 4a*). However, single-color events did impact future informed

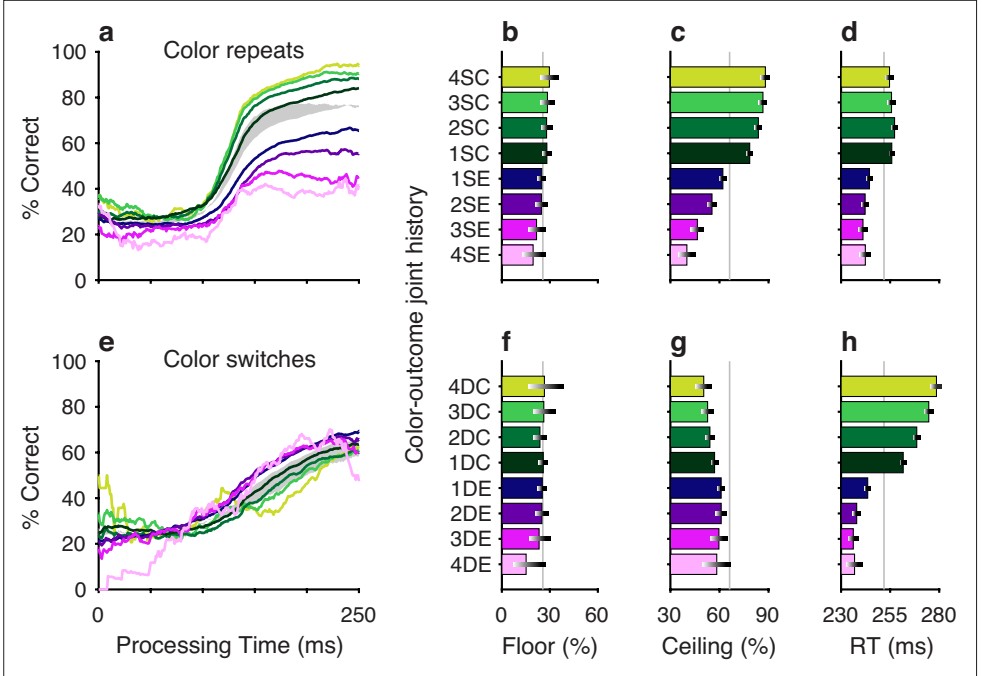

**Figure 5.** Perceptual accuracy is strongly modulated by the joint history of target color and trial outcome. (**a–d**) Task performance in color repeats. Data are conditioned on the histories of preceding trials having the same target color (S) and resulting in either correct (C, green spectrum; 1–4 preceding correct trials) or error outcomes (E, purple spectrum; 1–4 preceding error trials). (**e–h**) Task performance in color switches. Data are conditioned on the histories of preceding trials having a different (D) target color and resulting in either correct (C, green spectrum; 1–4 preceding correct trials) or error outcomes (E, purple spectrum; 1–4 preceding error trials). Formatting for the tachometric curves (**a, e**), floor accuracies (**b, f**), ceiling accuracies (**c, g**), and mean reaction times (RTs) (**d, h**) is the same as in *Figure 2*. The gray shaded areas in a and e indicate ranges of history effects obtained without taking outcome history into account, for color repeats (from 1S–4S in *Figure 2a*, blue curves) and color switches (from 1D–4D in *Figure 2a*, red curves), respectively.

choices, as indicated by the ceilings of the same curves (*Figure 4b*). The impact of a color repeat (*Figure 4b*, S conditions, blue circles) versus a switch (*Figure 4b*, D conditions, brown circles) was fairly symmetric about a mean performance baseline and accounted for a difference of approximately 13 percentage points in the likelihood of making a correct informed choice on the trial immediately following the reference event (1S versus 1D). This color bias decayed within 5 trials or so; it had a demonstrable influence over informed choices for the next 4 trials (*Figure 4b*, 1S4x versus 1D4x; note separation between 95% CIs) but largely faded thereafter. Such time course is highly consistent with prior measurements based on nonurgent versions of the oddball task (*Bichot and Schall, 2002*).

Notably, though, the inter-trial temporal dynamic for location bias (*Figure 4c and d*) was markedly different from that of color. In this case, the biasing effect of a location event, whether a repeat (S conditions) or a switch (D conditions), was smaller overall, producing a maximum difference of 7 percentage points in ceiling accuracy (*Figure 4*, 1S versus 1D). More importantly, the location bias was evident for both uninformed (*Figure 4c*, floor) and informed choices (*Figure 4d*, ceiling) and showed greater persistence than the color bias. In this case, a single event had a significant influence up to 9 trials into the future (*Figure 4c, d*, 1S8x versus 1D8x; note separation between 95% CIs).

## Color- and location-driven biases are strongly modulated by outcome

Goal-directed behavior is, by definition, sensitive to outcome (*Wolfensteller and Ruge, 2012*; *Dolan and Dayan, 2013*). When an action aims to achieve a particular objective (e.g., looking toward the color singleton to obtain reward), the result influences subsequent behavioral strategy. But in addition, as we show here, it also has powerful effects on the development of nonstrategic perceptual and motor-based selection history biases.

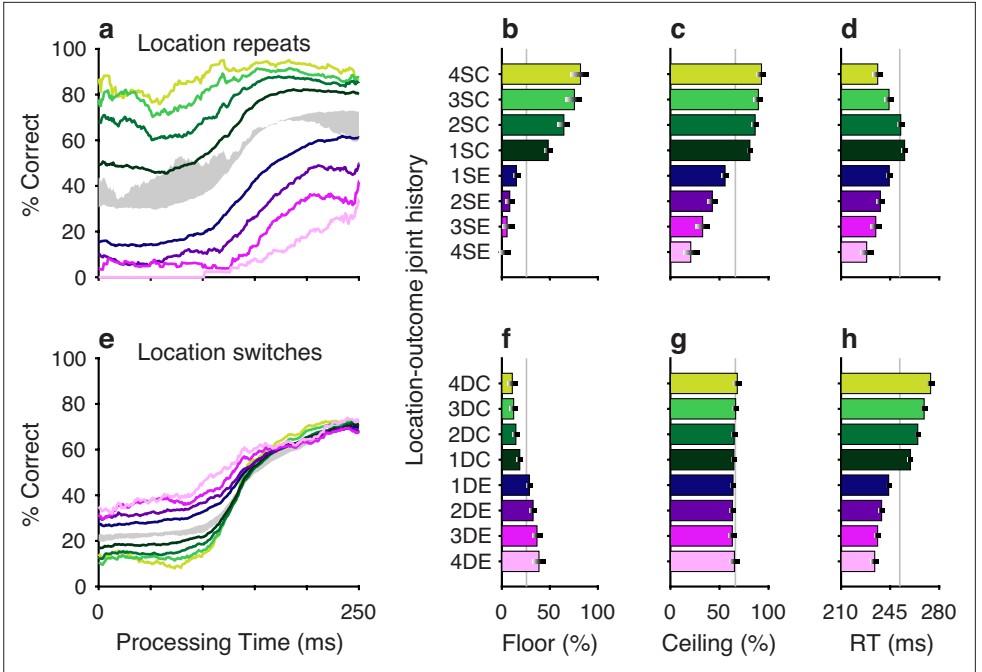

**Figure 6.** Motor bias is strongly modulated by the joint history of target location and trial outcome. The format of this figure is exactly the same as that of *Figure 5*, except that it considers the joint history of target location and trial outcome (rather than target color and trial outcome). The gray shaded areas in a and e indicate ranges of history effects obtained without taking outcome history into account, for location repeats (from 1S–4S in *Figure 2e*, blue curves) and location switches (from 1D–4D in *Figure 2e*, red curves), respectively.

The impact of outcome (reward versus no reward) on perception was quite apparent in the way that it modulated the expression of history-induced color bias (*Figure 5*). Tachometric curves conditioned on previous trial repetitions of either the same (*Figure 5a*) or different (*Figure 5e*) target color were jointly conditioned on outcome; i.e., different curves were generated depending on whether the color repetitions were associated with correct (*Figure 5*, green color spectrum) or incorrect (*Figure 5*, magenta color spectrum) choices. The biasing impact of the outcome was large. For example, for tachometric curves in which each choice was preceded by four trials of the same target color, we observed a difference of nearly 50 percentage points in ceiling performance depending on whether the previous same-color trials were correct or incorrect (*Figure 5a*, light green versus light magenta traces; *Figure 5c*, 4SC versus 4SE). Thus, for repeats of the same target color, a history of successful outcomes increased the likelihood of correctly choosing the target, whereas a history of unsuccessful outcomes decreased this likelihood. In other words, the same stimulus sequences led to vastly different performance depending on their association with reward. For repeats of the alternative target color, the effect of outcome on ceiling performance was more subtle but qualitatively consistent (*Figure 5e and g*). In this case, when the different target color was associated with success on previous trials, choosing the current target color was less likely (*Figure 5e and g*, green); and vice versa, when the different target color was previously associated with errors, choosing the current target color was comparatively more likely (*Figure 5e and g*, magenta).

The impact of outcome on location bias was similarly strong (*Figure 6*). When the same target location was repeated, the outcome either reinforced or diminished the likelihood of choosing the target, depending on whether the repeats were associated with correct (*Figure 6a–c*, green) or incorrect (*Figure 6a–c*, magenta) choices. Conversely, repeated occurrences of different target locations yielded the complementary result; for these, correct choices diminished the likelihood of choosing the current target location (*Figure 6e–g*, green) and errors increased this likelihood (*Figure 6e–g*, magenta). As expected, these outcome-dependent variations in motor bias were most evident for uninformed choices, as indicated by modulation of the floor values (*Figure 6b and f*) of the conditioned tachometric curves (*Figure 6a and e*).

The effects of unrewarded (error) outcomes on accuracy are consistent with the idea that these expressions of bias were not the product of a deliberate strategy to optimize behavior. This is true for repetitions of both the same target color and the same target location. For example, if the color singleton was red on 4 consecutive trials and the subject made an error (i.e., looked to a green distracter) on each of those trials, the results suggest a progressive perceptual devaluation of red stimuli (*Figure 5a and c*, magenta). This, in turn, suggests that the attribution of outcome on each trial was to the color of the singleton (e.g., red), not the color of the distracter (e.g., green) that was erroneously targeted by the saccade – an attribution that ignores the role of the motor act in causing the outcome. A similar logic applies to repeated target location. If the target repeatedly appeared in the same location and was missed (i.e., the subject looked elsewhere), the subject was less likely to select the target at that location on the current trial (*Figure 6a–c*, magenta). Again, to the extent that error history biased behavior, the attribution of the unrewarding outcome appears to be tied to the singleton location, not the erroneous motor act.

A common trend for the influence of outcome on RT was observed for both color and location bias, and for same and different target repetitions within each of these feature dimensions. Specifically, RTs tended to be shorter when preceded by one or more error trials (*Figures 5d, h, 6d and h*, magenta) and longer when preceded by correct trials (*Figures 5d, h, 6d and h*, green). This effect on RT is consistent with a general association between time and accuracy that, by design, is particularly acute in urgent choice tasks; namely, long RTs typically lead to long PTs and thereby to high accuracy, whereas short RTs typically lead to short PTs and thereby to near-chance performance. In general, however, relatively modest differences in mean RT, like those between SC and DC conditions (green bars in *Figures 5d, h, 6d and h*), are difficult to attribute because variations in motor urgency (and thus RT) can be very large even under relatively simple conditions, such as during saccades to lone visual targets (*Hikosaka et al., 2006*; *Hauser et al., 2018*). Moreover, in an urgent choice paradigm like the CO task, numerous mechanisms are likely to determine the RT distribution (*Shankar et al., 2011*; *Goldstein et al., 2022*; *Oor et al., 2023*). As discussed below, while differences in choice accuracy were slight across the two monkey subjects, their differences in mean RT were larger, consistent with the notion that, in general, RTs have more complex variability.

## Little evidence of interaction between color and location histories

Up to now, the results indicate that both the history of target colors and the history of target locations produce strong biases on task performance that are heavily gated by the outcomes (success/failure) of the trials within those histories. Although the resulting color- and location-driven effects have distinct temporal signatures, it is unclear whether the two sources of bias interact or, alternatively, act independently to determine performance at long PTs (>150 ms), when motor and perceptual mechanisms must coordinate and overlap. To assess the functional interaction between color- and location-driven biases, as well as its dependence on outcome, we devised an analysis in which the individual, separate effects of two trial histories on performance (during informed trials) are compared to their joint, combined effect (Methods; *Salinas and Stanford, 2024*).

In this approach, predictability serves as a criterion for interaction. Take event $C$ to represent the outcome of an informed trial, which can be correct ($C = 1$) or incorrect ($C = 0$), and take $A$ and $B$ to represent two history sequences for different variables, say one for target color and another for target location. Then, $P(C|A, B)$ represents the probability of an outcome given that histories $A$ and $B$ were observed jointly. The main idea is to estimate this probability in two ways: (1) by calculating it directly from the experimental data, and (2) by predicting it based on the separate probabilities associated with the two histories, $P(C|A)$ and $P(C|B)$, assuming no statistical interaction between the history variables. If the prediction agrees with the measurement, the natural interpretation is that the two history variables exert independent influences on performance. Conversely, a significant discrepancy is interpreted as evidence of a functional interaction between the two history variables with respect to task performance.

This is best understood with an example. First, from the data, we calculate the probability of a correct outcome given a combined sequence; say, that the target in the prior trial had a different color *and* the same location. The measured value is $P(C = 1|DS) = 0.68$ in this case. Then, we calculate the outcome probabilities given the two separate histories, namely $P(C = 1|D$ color), which is the probability correct given that the prior target color was different, and $P(C = 1|S$ location), which is the

probability correct given that the prior target location was the same. The values from the experimental data are 0.59 and 0.71, respectively. Then, by assuming a form of functional independence (conditional independence; Methods; *Salinas and Stanford, 2024*) between the two history variables, color and location, it is possible to use these two values measured separately to predict the probability correct when the two history conditions occur jointly. The predicted probability correct is 0.64 in this case (using *Equation 7* and the fact that the overall probability correct was 0.66). The discrepancy between the two resulting numbers (0.68 versus 0.64) is small, so in a plot of predicted versus measured accuracy, they correspond to a point near the diagonal (*Figure 7a*, top row, point marked 1DS).

When the same analysis is performed for all the possible sequences that combine color and location going one trial back (sequences 1DD, 1SD, 1DS, 1SS), the four data points lie fairly close to the diagonal (*Figure 7a*, top row), suggesting that the interaction between the two history variables is, at most, weak. To obtain a more complete characterization of the interdependence between color and location histories, the analysis is extended to sequences that go back further in time, up to 4 trials before the trial being evaluated (*Figure 7a*, $H = 1$ to $H = 4$, from top to bottom row). As the history length increases, more sequence combinations arise, and the contrast between predicted and measured accuracies includes more data points (with the number of trials represented by each point decreasing accordingly). For instance, going back 2 trials into the past, the 4 possible color sequences (SS, SD, DS, DD) can be combined with 4 possible location sequences to yield 16 unique combinations of color and location, so 16 data points (*Figure 7a*, $H = 2$). But critically, the pattern that emerges is similar for all the history lengths: the data generally lie along the diagonal, with regression slopes close to 1 (*Figure 7a*, blue lines and $\beta_2$ coefficients). This indicates that, to a first approximation, color and location histories influence ceiling performance independently of each other.

To validate this analysis method, we applied it to a different pair of history variables, target color, and trial outcome. A strong interaction was expected for this combination of variables based on the earlier results, which showed that the same color history can either increase or decrease the ceiling accuracy depending on the outcome of past trials considered (*Figure 5*, green versus magenta). In this case, the effect of a given color history was paired with the effect of a given success history. For instance, $P(C = 1|1D$ color$)$, which is the same as above, was now paired with $P(C = 1|1E$ outcome$)$, the probability that the outcome of an informed trial is correct given that the prior outcome was an error, to estimate the joint effect of the two histories; i.e., the probability $P(C = 1|1DE)$, which is the probability correct when in the preceding trial the target color was different *and* the choice was erroneous (*Figure 7b*, top row, point marked 1DE). For this example, the discrepancy between predicted (0.54) and empirical (0.62) probabilities is larger, and the four data points for the 1-trial histories deviate further from the diagonal (*Figure 7b*, $H = 1$, top row). Now, as longer histories are considered, the pattern that emerges demonstrates a consistent deviation from the diagonal and a shallower slope (*Figure 7b*, blue lines and $\beta_2$ coefficients), indicating that the predicted ceiling accuracy typically underestimates the magnitude of the joint history effect: when the empirical accuracy is above the mean (>66% correct along the $x$ axis), the prediction is typically too low, and when the empirical accuracy is below the mean (<66% correct along the $x$ axis), the prediction is typically too high. Thus, as expected, for color and outcome histories, the results of this independence analysis are consistent with an underlying functional interaction.

Very similar results were obtained when we considered the combined effect of target location and outcome histories, except that the discrepancies between predicted and measured probabilities were slightly larger than for color and outcome (*Figure 7c*, blue lines and $\beta_2$ coefficients). Again, this is consistent with earlier results (*Figure 6*) and denotes a strong functional interaction, in this case between outcome and target location histories.

Finally, we considered again the interaction between target color and target location histories, but this time pairing each of them with success history. In this case, the goal was to predict the probability of a correct outcome given the recent history of the three variables. For example, we measured $P(C = 1|1SSC)$, which is the probability that a choice is correct when, in the preceding trial, the target color was the same, the target location was the same, and the outcome was correct. Then, we compared the measured probability with an estimate based on two quantities, $P(C = 1|1SC$ color$)$, which is the probability correct given that the prior trial had the same color and was correct, and $P(C = 1|1SC$ location$)$, which is the probability correct given that the prior trial had the same location and was correct. For this example, the measured and predicted probabilities were nearly identical

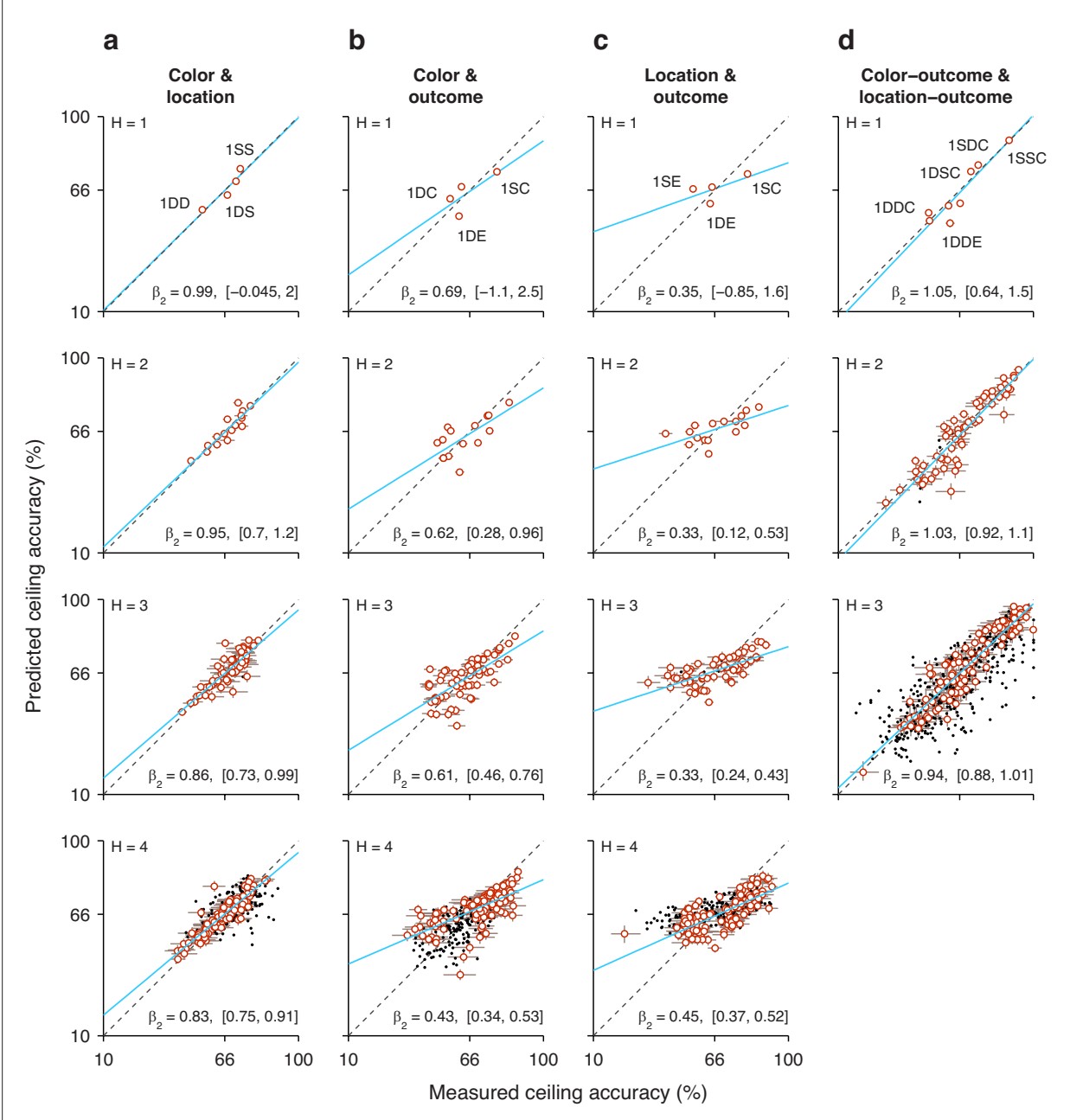

**Figure 7.** Tests of interaction between individual histories. Each panel plots the predicted task performance at long processing times (PTs) (ceiling accuracy) based upon independent contributions from two history effects (*y* axis), contrasted with the actual, measured task performance (*x* axis). Rows correspond to comparisons based on histories going 1, 2, 3, or 4 trials back (H, from top to bottom). Each point represents one specific history sequence. Labels in top panels identify a few combined history sequences in each case. Error bars represent 95% confidence intervals (CIs). Red and black identify individual points with CI spans smaller (red) or larger (black) than 15 percentage points, which divides the data into high- and low-reliability points. Regression lines based on the red points are plotted in blue, with the slope ($\beta_2$) and its 95% CI indicated in each case. Dotted diagonal lines indicate equality. Note that the overall probability of success across all trials is 66% correct. (**a**) Interaction between target color and target location histories. (**b**) Interaction between target color and outcome histories. (**c**) Interaction between target location and outcome histories. (**d**) Interaction between target color and target location histories, each combined with outcome. Note that the joint effect of color and location histories is highly predictable based on their individual effects on performance.

(0.892 versus 0.897; *Figure 7d*, top row, point marked 1SSC). There were 8 unique sequences going back 1 trial, and most of the corresponding data points fell close to the equality diagonal (*Figure 7d*, $H = 1$, top row). For longer sequences, the resulting data points spanned a wide range of accuracies (for $H = 3$, going from <20% correct to almost 100% correct), but the alignment was consistent with a slope of 1 in all cases. The data show that color- and location-driven biases exert largely independent effects on ceiling performance, especially once outcome history is taken into consideration.

In summary, the results from our analyses based on predictability are consistent with the notion that although both color and location histories interact strongly with outcome history, the resulting color- and location-driven biases make approximately independent contributions to performance during informed choices.

## Selection history effects are consistent across monkeys

Considering the diverse sources of history-driven biases that impact performance in the CO task, it would not be surprising if selection history effects were to manifest to widely varying degrees across individual subjects. Interestingly, though, we found that while RTs indeed tended to be somewhat

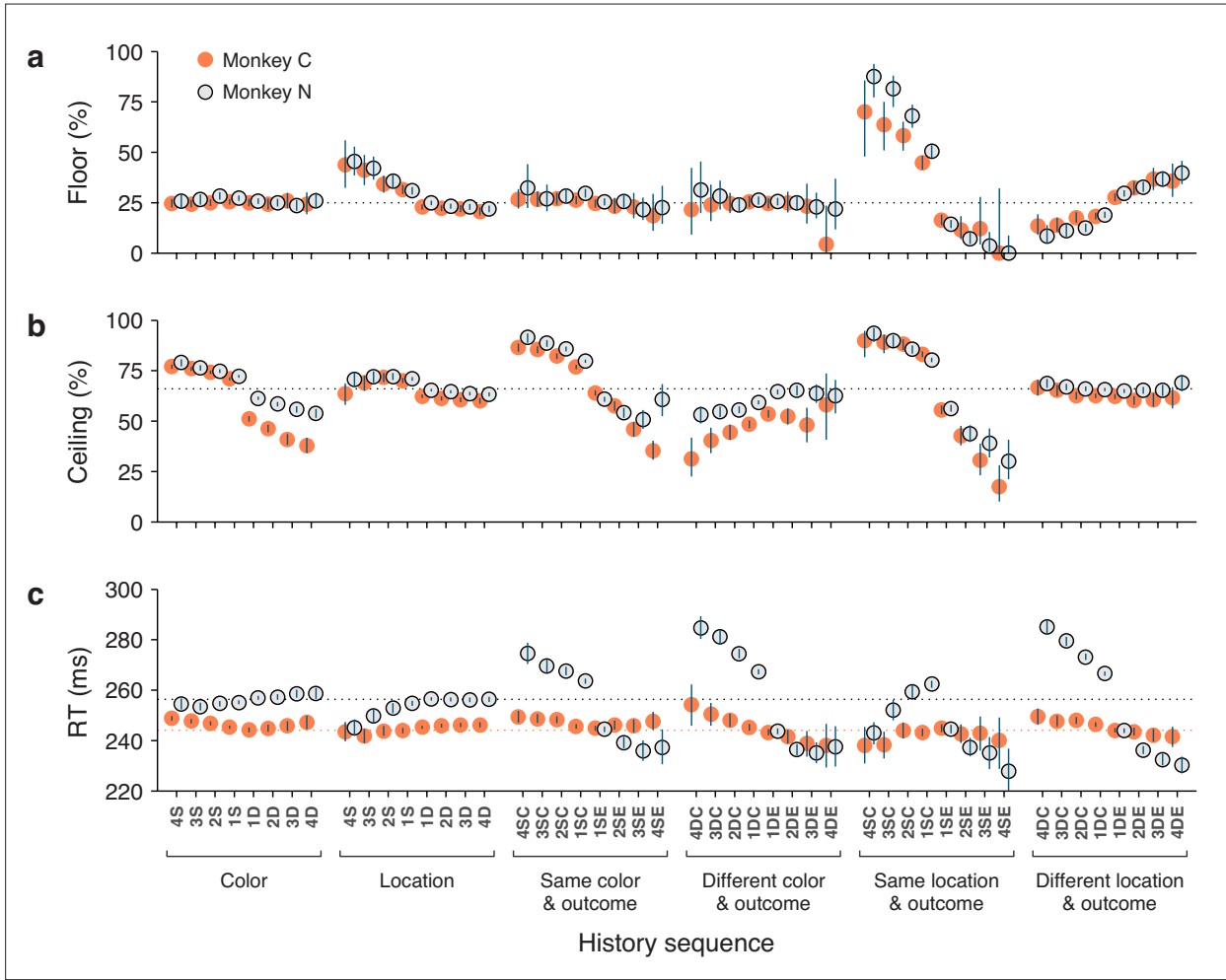

**Figure 8.** Consistency of history effects across individual subjects. The data in this figure are analogous to the bar plots shown in *Figures 2, 5, and 6*, but for each of the two monkeys. In each panel, the x axis indicates specific history sequences (see labels in c). On the y axes, orange filled circles are results from monkey C and open circles are from monkey N. For clarity, the two sets of data points are slightly offset from each other along the x axis. Blue error bars indicate 95% confidence intervals (CIs) across trials. (**a**) Choice accuracy (percent correct) in short-processing time (PT) trials (PT <100 ms, floor) conditioned on each history sequence. The dotted line marks chance performance (25% correct). (**b**) Choice accuracy (percent correct) in long-PT trials (PT >150 ms, ceiling) conditioned on each history sequence. The dotted line marks average performance for informed choices (66% correct). (**c**) Mean reaction time (RT) conditioned on each history sequence. Data points include all trials in each condition (correct and incorrect, all PTs). The dotted lines mark the overall mean RTs for the two monkeys (with the RTs of monkey N shifted by 29 ms; Methods).

idiosyncratic, variations in choice accuracy across history conditions were highly consistent in both sign and magnitude across the two monkey subjects.

Data from monkeys C and N were compared side-by-side for the history conditions already discussed in the preceding sections. For choice accuracy, the results recapitulated the trends identified earlier. During uninformed choices (*Figure 8a*), color history had essentially no impact, whereas saccades were strongly biased toward prior target locations, especially when they were associated with success (or reward). Across subjects, the corresponding modulations of the floor of the tachometric curve were nearly identical. During informed choices (*Figure 8b*), the effect of location history alone was more subtle than that of color alone, but both dimensions had a remarkably strong impact on performance when prior outcomes were taken into account. Across subjects, the corresponding modulations of the ceiling of the tachometric curve were again similar in this case, except that monkey C was somewhat more sensitive to switches in target color (*Figure 8b*; note slightly larger deviations from average in orange points for D color conditions).

In contrast to choice accuracy, the mean RTs were different across monkeys in several respects. First, as noted earlier, monkey N generally responded more slowly (*Figure 1d*). This was visible in the data (*Figure 8c*, note difference between dotted lines) even after having downshifted all the RTs of this subject (by 29 ms; Methods). And second, the magnitudes of the history-driven modulations in RT were considerably larger for monkey N than for monkey C when trial outcome was taken into account (*Figure 8c*, histories conditioned on outcome). Interestingly, however, for each type of history condition, the trends in these modulations were not dissimilar: for the D sequences, for instance, the changes across repetitions and between correct and error conditions were qualitatively the same for the two animals (*Figure 8c*, histories conditioned on different color and outcome, and different location and outcome).

These findings are in line with the notion that RTs are the product of numerous interacting factors, including sensory, cognitive, and motor mechanisms. They also illustrate a key property of the tachometric curve, which is that it is minimally sensitive to variations in urgency or motor preparation (*Stanford et al., 2010*; *Shankar et al., 2011*; *Seideman et al., 2018*; *Stanford and Salinas, 2021*). This is because, when trials are parsed according to PT, the variability in performance due to urgency is drastically curtailed. As a consequence, derived measures of choice accuracy, including the floor and ceiling values, are largely indicative of sensory and cognitive processing but not motor processing. Therefore, it is not surprising to find that the two monkey subjects showed relatively large differences in their sensitivities to history when measured by mean RT, but showed only slight differences when measured by choice accuracy.

## Discussion

When searching for a color oddball, the performance of trained monkeys is excellent as long as the color of the target remains fixed (*Bichot and Schall, 1999*; *Bichot and Schall, 2002*; *McPeek and Keller, 2004*; *Scerra et al., 2019*; *Figure 1—figure supplement 1c and d*). Otherwise, when target and distracter colors are randomly swapped across trials, accuracy drops precipitously (*Song and McPeek, 2015*; *Figure 1b*). Here, we investigated why. As in prior studies (*Busse et al., 2011*; *Gupta et al., 2024*), a large lapse rate (~35% in our case) that could be naively interpreted as stochastic behavior due to internal noise sources turned out to correspond, instead, to the manifestation of short-term biases driven by the recent history of events preceding each choice. In contrast to prior studies, we examined this problem under high urgency, which means that (uninformed) motor plans must be initiated early on. We exploited the temporal resolution afforded by the urgent-choice design to dissociate bias mechanisms that may start acting before the relevant sensory information becomes available from those that must start acting afterward. That is, mechanisms that bias the initial motor selection process versus mechanisms that bias the later perceptual evaluation of sensory features (color) and the subsequent choice. With this approach, we found that the histories associated with target location and target color made large contributions to choice accuracy that (1) were associated with motor and perceptual mechanisms, respectively, (2) had distinct time constants, (3) in both cases depended strongly on the success of past trials, and (4) to a large degree acted independently of each other during perceptually informed trials. Having teased apart these selection history biases, we found that the magnitude of their effect on visuomotor performance could be remarkably large.

History-driven biases are ubiquitous, but in the context of visuomotor choice tasks, or sensory-guided tasks more generally, they are typically observed when the relevant perceptual information is weak and the choice difficult. When associated with threshold performance in perceptual tasks, such effects are often referred to as 'serial dependencies' (*Kiyonaga et al., 2017*; *Manassi et al., 2023*; *Cicchini et al., 2024*). Many of these effects are specifically related to sensory representations kept in working memory, such as when a saccade is made to the location of a remembered visual cue (*Papadimitriou et al., 2015*; *Papadimitriou et al., 2017*), or during discrimination tasks in which two stimuli are presented sequentially, one after the other (*Akrami et al., 2018*; *Boboeva et al., 2023*). In other cases, the categorization of stimuli requires comparison with respect to an implicit boundary that must be represented internally, and the dependence on past trials is consistent with a boundary representation that is automatically updated as new stimuli are experienced (*Ashourian and Loewenstein, 2011*; *Raviv et al., 2012*; *Hachen et al., 2021*; *Mendonça et al., 2020*; *Sheehan and Serences, 2022*; *Boboeva et al., 2023*). In these types of studies, the data can be generally understood as perception resulting from current sensory evidence being combined with or compared to prior information stored internally, and the observed biases are most obvious when the available sensory evidence is weak and unreliable, so prior information takes precedence. In this framework, history-dependent perceptual biases are the manifestations of involuntary mechanisms for updating the relevant prior information (*Lak et al., 2020*; *Mendonça et al., 2020*). Such mechanisms are most useful under natural conditions, when statistical regularities are present (*Kiyonaga et al., 2017*; *Fritsche et al., 2020*; *Hermoso-Mendizabal et al., 2020*), but they may be counterproductive during typical laboratory tasks in which trials are uncorrelated (*Akrami et al., 2018*).

As in the studies just discussed, the task used here is a simple visuomotor paradigm – and yet, the differences are stark. In those studies, variations in performance were primarily driven by variations in stimulus strength or discriminability, and prior history typically led to relatively small deviations in accuracy of a few percentage points (but see *Mendonça et al., 2020*). Here, in contrast, the variability in performance was mainly driven by the amount of time available for processing the visual cue, and the magnitude of the history effects was extremely large (e.g., *Figures 5 and 6*), spanning nearly the full range from 0% to 100% correct for certain conditions. This is even more remarkable given that the task had no short-term memory requirements other than the task rule itself, and that the discriminability of the stimuli was always high – a condition under which the effects discussed in the previous paragraph typically vanish. Clearly, the framework developed for the abovementioned studies, based on the update of task-relevant information held in short-term memory, does not apply in this case.

Rather, our results are better understood as the confluence of two phenomena that have been described before, and for which the neural underpinnings have been at least partially characterized in monkeys. On one hand, the motor bias we observed is highly consistent with experiments showing that monkeys make much faster and much more accurate saccades to spatial locations that have been recently associated with a large reward than to locations that have not (*Lauwereyns et al., 2002*; *Ikeda and Hikosaka, 2003*; *Hikosaka et al., 2006*; *Isoda and Hikosaka, 2008*; *Hauser et al., 2018*). Such effects are caused by the recent history of reward-driven reinforcement going back a few (1–15) trials and are entirely compatible with the endogenous deployment of attention to a specific location tagged by reward (*Maunsell, 2004*; *Hauser et al., 2018*). Thus, although such spatial effects have been obtained under more minimalistic conditions (i.e., during saccades to lone visual targets), all indications are that they correspond closely to the motor biases in the CO task. Interestingly, however, expression of the strong bias that we observed in favor of previously rewarded target locations is likely to depend on the urgent nature of the CO task, because under nonurgent but otherwise comparable conditions, target location repetitions had no discernible effect on accuracy and only a small influence on RT (*Bichot and Schall, 2002*). This contrast suggests that the suspicion mentioned in the Introduction was correct, namely, that the impact of motor biases on *informed* choices is greatly amplified and much more evident when motor activity is already ongoing before the target selection occurs.

On the other hand, the perceptual bias we observed is highly consistent with color priming, a well-known effect whereby the perceptual processing of a given color in one trial facilitates the processing of the same color in the next trial. Although priming effects are typically reported as differences in RT, the color bias in the CO task is most likely the same as the 'priming of pop-out' reported in human participants (*Maljkovic and Nakayama, 1994*; *McPeek et al., 1999*), which is characterized as an attention-focusing mechanism that builds up cumulatively (as in *Figures 2a–d and 3*), decays within

5–8 trials (as in *Figure 4b*), and is largely involuntary (consistent with permanence in spite of months of practice). As with the motor bias, neurophysiological studies again indicate that the phenomenology is entirely compatible with known attention mechanisms; i.e., the priming results can be explained by short-lived adaptation in the responses of identified neural circuits in visual and oculomotor areas, which convey information about object salience and thereby guide the selection of saccade targets (*Westerberg and Schall, 2021*). In contrast to the motor bias, the color priming effect seems to be minimally affected by urgency, because its impact on choice accuracy under nonurgent conditions (*Bichot and Schall, 2002*) was comparable to that found here.

Two observations suggest that spatial and feature-based selection history effects are highly robust and hence relevant to real-world situations. First, the qualitative similarity just discussed between such effects as found in the CO task and in simpler, nonurgent tasks. And second, the fact that they are not strategic, on average neither helping nor hindering overall performance. Importantly, though, these are just two types of history-driven bias.

History-based influences on performance take many forms and probably involve a wide diversity of neural mechanisms and circuits (*Yashar et al., 2017*; *Failing and Theeuwes, 2018*; *Jiang and Sisk, 2019*; *Theeuwes, 2019*; *Anderson et al., 2021*). Consistent with this notion, we found little evidence of a functional interaction between the location- and feature-based biases (*Figure 7a and d*), in agreement with a prior psychophysical study (*Di Pace et al., 1997*). Also, although both serial dependencies and color priming are history-driven and perceptual in nature, they likely represent mechanistically distinct phenomena. Indeed, a study that directly compared them found that their effects were uncorrelated and concluded that they are mediated by separate mechanisms (*Galluzzi et al., 2022*). History effects are relevant not only to attention and visuomotor behaviors, as in the present study, but also to other cognitive functions such as working memory (*Papadimitriou et al., 2015*; *Papadimitriou et al., 2017*; *Kuo, 2016*; *Akrami et al., 2018*; *Boboeva et al., 2023*), value assessment (*Failing and Theeuwes, 2018*; *Constantinople et al., 2019*), or task switching (*Wylie and Allport, 2000*; *Monsell, 2003*; *Stoet and Snyder, 2007*). Given this, and the remarkable degree to which the performance of our monkey subjects was swayed by history-driven biases, we conclude that the contribution of recent history to behavioral variability is likely much larger than is generally appreciated, particularly under more naturalistic, less constrained conditions during which multiple such effects may be expressed (*Kattner et al., 2025*). This is an important topic for further investigation.

## Methods

### Subjects

All surgical and behavioral recording procedures were conducted in compliance with the Wake Forest University School of Medicine (WFUSM) Institutional Animal Care and Use Committee (IACUC, protocol A22-090), the National Institutes of Health Guide for the Care and Use of Laboratory Animals, and all relevant USDA regulations. Data were obtained from two purpose-bred adult male rhesus monkeys (*Macaca mulatta*, 7 years of age), both of which met WFUSM veterinary standards for health quality upon procurement. Daily health status was monitored by veterinary and lab staff throughout the duration of the study. Animals were housed in Allentown quad format cages, which met all regulatory requirements, in rooms equipped with a controlled light-dark cycle. Behavioral enrichment was provided by laboratory staff as per a WFUSM environmental enrichment policy. Feeding was ad libitum and provided to each animal's home cage by WFUSM Animal Resources Program staff. Supplemental food enrichment was provided daily by laboratory staff.

Monkeys N and C participated in a previous study performing a similar urgent choice paradigm paired with recordings from the superior colliculus, thus the current experiment utilized existing surgical preparations. Standard sterile surgical techniques under general anesthesia were employed for securing an MRI-compatible head post and an MRI-compatible recording cylinder to the skull (*Wyder et al., 2003*). Postsurgical pain was managed with the opioid analgesic buprenorphine hydrochloride delivered intramuscularly in 0.01–0.02 mg/kg doses, once perioperatively, followed by doses in 8–12 hr increments for the following 24–48 hr (unless otherwise specified by veterinary staff). Postsurgical inflammation was managed with the nonsteroidal anti-inflammatory drug ketoprofen delivered intramuscularly in 2.5–5.0 mg/kg doses, once perioperatively, followed by once daily administration for 2 days (unless otherwise specified by veterinary staff). After a minimum recovery

period of 2 weeks, positive reinforcement was used to train willingness to tolerate head restraint and, subsequently, to perform behavioral tasks involving saccadic eye movements to visual stimuli appearing on a computer monitor.

## Behavioral recording techniques

Standard operant methods with positive reinforcement were used to train monkey subjects on the behavioral task. For training and experimental sessions, liquid rewards for correct performance on visual tracking tasks served as the primary source of hydration. Animals were given the opportunity to work to satiety before returning to their home cage (satiety indicated by a decrease in motivation toward the later stages of a behavioral session). If necessary, supplementary liquid was provided in the animal's home cage to ensure adequate hydration as specified by WFUSM policy.

Behavioral tasks were performed with monkeys seated in a quiet, dimly lit room. Subjects maintained a comfortable upright seating position in a purpose-designed primate chair (Crist Instrument Co.) with head restrained to maintain a stable, straight-ahead orientation with respect to the visual display. Visual stimulus arrays consisted of gray, red, and green filled circles measuring 2.3° of visual angle in diameter and were presented on a VIEWPixx LED monitor (VPixx Technologies Inc, Saint Bruno, Quebec, Canada; 1920 × 1200 screen resolution, 120 Hz refresh rate, 12 bit color) at a viewing distance of 57 cm. Eye movements were sampled at 1000 Hz using an infrared camera-based eye tracking system (EyeLink 1000, SR Research, Ottawa, Canada). Stimulus presentation, behavioral task implementation, and data acquisition were controlled and coordinated using a custom-designed data acquisition system (Ryklin Software Inc).

## Behavioral task

The CO task (*Figure 1a*) is an urgent four-alternative choice paradigm in which subjects must make a saccade to a color singleton presented among three uniformly colored distracters (*Scerra et al., 2019*). Subjects begin by fixating a central gray spot on a black background. After 200 ms, a stimulus array consisting of four gray placeholders spaced 90° apart is presented to indicate the locations at which the singleton target and three distracters will appear (*Figure 1a*, Targets On). After a delay of 500–1000 ms, the fixation spot disappears (*Figure 1a*, Go), instructing the subject to make a saccadic choice within 450 ms for a chance at juice reward. Crucially, the relevant color information is withheld for a period of time ranging from 0 to 225 ms (*Figure 1a*, Gap), after which the gray placeholders change color to reveal the location of either a green target among three red distracters (*Figure 1a*, Cue) or a red target among three green distracters (not shown). Target identities and locations are randomly and independently assigned from trial to trial such that the prior probability of a color repetition is 0.5, and the prior probability of a location repetition is 0.25. A trial is correct if the subject makes an eye movement to the location of the oddball target within the allowable time window of 450 ms.

For monkey C, data were collected in the span of 20 months, in 79 experimental sessions that yielded 26,038 trials of the CO task. For monkey N, data were collected in the span of 21 months, in 105 experimental sessions that yielded 45,173 trials.

## Quantification and statistical analysis

All data analyses were performed in the MATLAB programming environment (R2020a; The MathWorks, Natick, MA, USA). Behavioral performance was quantified as detailed in prior reports (*Stanford et al., 2010*; *Scerra et al., 2019*; *Goldstein et al., 2022*; *Seideman et al., 2022*; *Oor et al., 2023*). RT was measured from the time of the Go signal to the onset of the saccade (when eye velocity had surpassed 50°/s). For each trial, the PT – the interval between Cue and saccade onset – was calculated by subtracting the duration of the Gap interval from the RT on that trial.

The psychophysical performance of each subject was evaluated using the tachometric curve (*Figure 1b*), a psychometric function that plots the probability of a correct choice as a function of PT, or cue viewing time (*Stanford et al., 2010*; *Scerra et al., 2019*; *Goldstein et al., 2022*; *Seideman et al., 2022*; *Oor et al., 2023*). Tachometric curves were generated by sorting the trials by PT and computing the fraction correct for all trials within each PT bin, with bins (50 ms width) shifting every 1 ms. The tachometric curves of both subjects (*Figure 1b*) show that performance varies from chance (probability correct = 0.25) to asymptotic with increasing PT. For PT < 100 ms, behavior is at chance,

and such trials are deemed 'uninformed'. For PT > 150 ms, performance rises rapidly to reflect the increasing likelihood that the choice is guided by color information, and these choices are considered 'informed'.

Separate tachometric curves were generated for each history sequence, and for each resulting curve, two key parameters were calculated: the percent correct for uniformed choices (or floor accuracy) and the percent correct for informed choices (or ceiling accuracy). The calculation of these quantities was independent of the bin size used to depict the tachometric curves: the floor simply corresponded to the percent correct for all history-conditioned trials with PT < 100 ms, whereas the ceiling corresponded to the percent correct for all history-conditioned trials with PT > 150 ms. For both quantities, 95% confidence intervals (CIs) were evaluated using binomial statistics, specifically, the Agresti-Coull method (*Agresti and Coull, 1998*). Significance levels were established via resampling methods (*Siegel and Castellan, 1988*; *Hesterberg, 2014*). Mean RT together with standard error of the mean (SEM) was also calculated for the dataset associated with each history sequence. Such history-conditioned mean RTs are based on both correct and incorrect trials at all PTs.

## Data pooling

The effects of trial history on performance were qualitatively similar for the two monkey subjects (see *Figure 2—figure supplement 1* and last section of Results), but their PTs were somewhat different. Thus, to pool the data across the two subjects, we first aligned their mean tachometric curves (*Figure 1b*). Optimal alignment was determined as follows. First, one of the tachometric curves was rescaled and shifted so that the average absolute difference between the tachometric curves from the two monkeys was minimized. The formula for the transformed curve is

$$z(j) = g\,(y_2(j + \delta) + b) \qquad (1)$$

where $y_2$ and $z$ are the original and transformed values (for fraction correct), respectively, $j$ indicates a PT bin, and the transformation parameters are the gain ($g$), baseline shift ($b$), and the shift along the PT axis ($\delta$). These parameters were set to minimize

$$\sum_j \left( \left| z(j) - y_1(j) \right| \right) \qquad (2)$$

which is the total deviation between tachometric curve 1 ($y_1$) and the transformed version of tachometric curve 2 ($z$). The variable of interest for the purposes of optimally aligning the two datasets was $\delta$, and it indicated that the tachometric curve for monkey N (*Figure 1b*, bottom) should be shifted leftward by 29 ms. Thus, before pooling the data from the two monkeys, we first subtracted 29 ms from all the RTs from monkey N, which in turn shifted all its PTs by the same amount. Although the applied shifts in the tachometric curve (*Figure 2—figure supplement 1e*, gray bar) and the RTs (*Figure 2—figure supplement 1h*, dark bars) of monkey N were in no way critical to the results, the temporal alignment in the pooled data reduced the variance of the measured quantities.

## Feature history

Trials were sorted post hoc based on the number of consecutive occurrences in which the target color on immediately preceding trials was either the same or different than that on the current trial. Because target history effects were qualitatively similar for red and green targets, trials were grouped into same (S) versus different (D) categories for subsequent analyses of the effects of target color history (*Figure 2—figure supplement 2*).

Trials classified as S were those preceded by $N$ trials with a target of the same color (e.g., *Figure 2a*, blue spectrum). The classification was such that trials preceded by at least 1 trial with a target of the same color were termed 1S trials (sequences red-**red** and green-**green**, where the bolded word indicates the trial of interest). Trials preceded by at least 2 trials with a target of the same color were termed 2S trials (red-red-**red** and green-green-**green**), etc. The S trials reveal the impact of color repetition.

Analogously, trials classified as D (e.g., *Figure 2b*, red spectrum) were those preceded by $N$ trials with a target of the opposing color, which was different. As such, trials preceded by at least 1 trial with a target of the opposite color were designated 1D trials (green-**red** and red-**green**), trials preceded

by at least 2 trials with a target of the opposite color were designated 2D trials (green-green-**red** and red-red-**green**), and so on. The D trials reveal the impact of a color switch.

### Location history

The orientation of the 4-stimulus array and the eccentricity of the stimuli were varied pseudorandomly across experimental sessions. For the purpose of subsequent analysis, each target location was assigned to one of four designated quadrants: right, up, left, and down. Progressing counterclockwise, the target location was categorized as follows: right, if corresponding to a positive value along the x axis or anywhere within the first (upper-right) quadrant; up, if corresponding to a positive value along the y axis or within the second (upper-left) quadrant; left, if corresponding to a negative value along the x axis or within the third (lower-left) quadrant; down, if corresponding to a negative value along the y axis or within the fourth (lower-right) quadrant.

Independently of feature history, trials were sorted based upon the number of consecutive occurrences in which the target location (i.e., quadrant) on immediately preceding trials was the same or different than for the current trial. Trials classified as S were those preceded by $N$ trials with a target in the same location (e.g., *Figure 2b*, blue spectrum). Trials preceded by at least 1 trial with a target in the same location were 1S trials (sequences right-**right**, up-**up**, left-**left**, down-**down**, where the bolded word again indicates the trial of interest). Trials preceded by at least 2 trials with a target in the same location were 2S trials (right-right-**right**, up-up-**up**, left-left-**left**, down-down-**down**), etc. Trials classified as D (e.g., *Figure 2b*, red spectrum) were those preceded by $N$ trials with a target in a different location. Trials preceded by at least 1 trial with a target of a different location were 1D trials (examples: up-**right**, down-**right**, left-**down**, right-**up**; 12 possible combinations), trials preceded by at least 2 trials with a target of (any) different location were 2D trials (examples: left-up-**right**, left-left-**up**, up-down-**left**, etc; 36 possible combinations), and so on. As with color, S and D trials reveal the impact of repeats and switches, but of target location in this case.

### Outcome history

Trials were additionally sorted based upon the number of correct or error trials that preceded the trial of interest. Trials preceded by at least 1 correct trial were designated 1C (sequence correct-x, where the x indicates the outcome of the trial of interest). A trial preceded by at least 2 correct trials was designated 2C (correct-correct-x), etc. The complementary classification scheme was applied for error trials such that trials preceded by at least 1 error were labeled 1E (error-x), while those preceded by at least 2 error trials were 2E (error-error-x), etc.

### Feature and location histories combined with outcome history

Utilizing the classification schemes described above, feature and location history effects were further conditioned on prior successes or failures. Here, the S and D sequences were separated into same/correct, same/error, different/correct, and different/error subgroups. Accordingly, trials immediately preceded by at least 1 successful trial with a target of the same color (or location) were designated 1SC trials; trials preceded by at least 2 correct trials with targets of the same color (or location) were considered 2SC trials, and so on (e.g., *Figure 5b*, green spectrum). Mirroring this analysis, S trials were conditioned on error history: trials preceded by at least 1 error trial with a target of the same color (or location) were designated 1SE trials, while trials preceded by at least 2 errors with targets of the same color (or location) were designated 2SE trials, etc. (e.g., *Figure 5b*, purple spectrum). This sorting structure was duplicated for the complementary set of D analyses, which produced sequences 1DC, 2DC, etc. and 1DE, 2DE, etc. for correct and error combinations, respectively (e.g., *Figure 5f*).

### Quantifying functional interactions between history variables

To understand and quantify the functional interactions between the three variables of interest, target color, target location, and trial outcome, we devised an analysis based on the concept of conditional independence (*Dawid, 1979*), which provides a useful characterization of how three stochastic variables may relate to each other. This widely applicable methodology is described in detail elsewhere (*Salinas and Stanford, 2024*), but the following is a brief overview.

Two events $A$ and $B$ are conditionally independent relative to a third event $C$ if their joint probability given $C$ is such that

$$P(A, B|C) = P(A|C)\, P(B|C) \tag{3}$$

where $P(A, B|C)$ is the probability that events $A$ and $B$ occur given that event $C$ has occurred. In other words, if $C$ is known, $A$ and $B$ occur independently of each other. An equivalent formulation of this idea is to say that, once condition $C$ is known, the probability that $B$ occurs is fixed, regardless of whether $A$ occurs or not. That is,

$$P(B|A, C) = P(B|C) \tag{4}$$

is an alternate definition. Note that these expressions are distinct from the standard notion of independence, which for events $A$ and $B$ would correspond to

$$P(A, B) = P(A)\, P(B). \tag{5}$$

This condition is different and irrelevant to the problem at hand. Conditional independence is essentially a constraint on the three-way relationships between the three variables of interest, as captured by their joint probability, $P(A, B, C)$. Here, it is useful because it provides a quantitative reference for the effect on $C$ that $A$ and $B$ are expected to have when considered individually versus when considered simultaneously.

In the current study, we identify event $C$ with the outcome of a trial (correct or incorrect) and events $A$ and $B$ with selection histories preceding that trial. For this application, we are interested in determining whether different histories (for target color, target location, or success) exert independent effects on trial outcome, or alternatively, whether they interact to reinforce each other (producing a stronger-than-expected effect) or interfere with each other (producing a weaker-than-expected effect). To generate a baseline expectation or benchmark for the no-interaction case, we consider how event $C$ depends on events $A$ and $B$. The quantity that we seek is

$$P(C|A, B) = P(A, B|C)\frac{P(C)}{P(A, B)} \tag{6}$$

where the right side is the standard identity based on Bayes theorem. When $A$ and $B$ are conditionally independent with respect to $C$, *Equation 3* can be used to factorize all the terms where $A$ and $B$ appear jointly on the right-hand side of *Equation 6*, and everything can be put in terms of $P(C|A)$ and $P(C|B)$, which describe how choice outcome depends on each of two history variables separately. If $C=1$ and $C=0$ indicate a correct and an incorrect choice, respectively, then the result of combining *Equations 3 and 6* is, after some algebra,

$$P(C=1|A, B) = \frac{P(C=1|A)P(C=1|B)}{P(C=1|A)P(C=1|B) + P(C=0|A)P(C=0|B)\dfrac{P(C=1)}{P(C=0)}} \tag{7}$$

(*Salinas and Stanford, 2024*). This expression represents a prediction based on three quantities: the effect of $A$ alone on choice outcome, the effect of $B$ alone on choice outcome, and the overall probability of a correct choice (i.e., the prior, $P(C = 1)$). It is a model-free benchmark for quantifying how much knowledge is gained when two sources of evidence are considered together rather than separately – if they do not interact. This prediction applies to any pairwise combination of separate history terms that depend on $A$ and $B$, and is to be contrasted with the empirical result for the probability of a correct choice given that $A$ and $B$ are known *simultaneously*.

For example, suppose that the following numbers are measured from the data: (1) the probability of a correct choice given that the oddball color in the preceding trial was the same as in the current trial

$$P(C = 1|1\text{S color}) = 0.80 \tag{8}$$

(2) the probability of a correct choice given that the preceding outcome was correct

$$P(C = 1|1\text{C outcome}) = 0.75 \tag{9}$$

and (3), the overall probability of a correct choice

$$P(C = 1) = 0.70 \tag{10}$$

Then, compute the expected probability of success given the joint condition,

$$P(C = 1|1S \text{ color, } 1C \text{ outcome}) \tag{11}$$

by inserting the above values into *Equation 7*. The result is 0.84; this is the expectation for the probability of a correct choice when both the target color and outcome histories are known. It is larger than either of the probabilities conditioned on a single history variable because they are in agreement (both above the prior) and because together they are more informative of choice outcome. This prediction is to be contrasted with the actual data, i.e., with the probability $P(C = 1|1SC)$ obtained by considering all the trials in which the preceding trial was both correct and of the same target color. Any significant discrepancy from the prediction is indicative of an interdependence between color and outcome histories with respect to their impact on the outcome of the current choice.

This analysis was used to uncover functional interactions between target color, target location, and outcome histories in various pairwise combinations (*Figure 7*). To obtain a complete description of how each pair of history variables influences the current choice, this procedure was applied to all the possible history sequence pairs going from 1 to 4 trials back, with each pair providing a predicted value (based on limited data and the conditional independence assumption) and an empirical value (based on the full dataset).

## Time course of history effects

The effect of a single past event fades with time, i.e., with the number of subsequent trials, and to estimate the corresponding time constant, we designed an analysis that runs as follows. First, consider trial sequences in which the color of the target was the same for the current trial and for the $N$th trial back, but regardless of what happened in the trials in between. Thus, for $N=1$ we have the 1S case mentioned earlier, which includes sequences red-**red** and green-**green**, where the bolded word indicates the trial of interest; for $N=2$ we have the case 1S1x, which includes sequences red-x-**red** and green-x-**green**, where the x indicates a trial with either target color, red or green; for $N=3$ we have the case 1S2x, which includes sequences red-xx-**red** and green-xx-**green** with two intervening trials of either color; for $N=4$ we have the case 1S3x, which includes sequences red-xxx-**red** and green-xxx-**green**; and so on. By plotting the probability of a correct choice for each one of these conditions as a function of $N$, the effect of a single S stimulus can be tracked as a function of time, i.e., trial number (*Figure 4a and b*, blue data).

The same procedure can be applied to reveal the time course of the modulatory effect caused by a single D stimulus (*Figure 4a and b*, brown data). In that case, one would consider the sequences labeled as 1D (green-**red** and red-**green**), 1D1x (green-x-**red** and red-x-**green**), 1D2x (green-xx-**red** and red-xx-**green**), 1D3x (green-xxx-**red** and red-xxx-**green**), and so forth. Together, the time courses for the S and D cases measure how rapidly the information about target color dissipates as further trials of the task are performed. Entirely analogous procedures were used to determine the time courses corresponding to same or different target locations, which also involved sequences labeled 1S, 1S1x, 1S2x, 1S3x, and so on (*Figure 4c and d*, blue data), and 1D, 1D1x, 1D2x, 1D3x, and so on (*Figure 4c and d*, brown data).

## Acknowledgements

We thank Denise Anderson for technical and administrative support. Research was supported by the National Institutes of Health through grant R01EY025172 (to ES and TRS) from the National Eye Institute. The funders had no role in study design, data collection and interpretation, or decision to publish.

## Additional information

### Competing interests

Emilio Salinas: Reviewing editor, eLife. The other authors declare that no competing interests exist.

## Funding

| Funder | Grant reference number | Author |
|---|---|---|
| National Eye Institute | R01EY025172 | Emilio Salinas<br>Terrence R Stanford |

The funders had no role in study design, data collection and interpretation, or the decision to submit the work for publication.

## Author contributions

Emily E Oor, Data curation, Formal analysis, Investigation, Visualization, Writing – original draft; Emilio Salinas, Conceptualization, Resources, Software, Formal analysis, Supervision, Funding acquisition, Validation, Investigation, Methodology, Project administration, Writing – review and editing; Terrence R Stanford, Conceptualization, Resources, Supervision, Funding acquisition, Investigation, Visualization, Methodology, Writing – original draft, Project administration, Writing – review and editing

## Author ORCIDs

Emily E Oor ⓘ https://orcid.org/0000-0003-1276-5281
Emilio Salinas ⓘ https://orcid.org/0000-0001-7411-5693
Terrence R Stanford ⓘ https://orcid.org/0000-0003-0759-5599

## Ethics

This study was performed in strict accordance with the recommendations in the Guide for the Care and Use of Laboratory Animals of the National Institutes of Health. All of the animals were handled according to approved institutional animal care and use committee (IACUC) protocol A22-090 of the Wake Forest University School of Medicine. Standard sterile surgical techniques were performed under general anesthesia and post-surgical pain was managed with opioid analgesic to minimize pain and distress.

Reviewer #1 (Public review): https://doi.org/10.7554/eLife.100280.3.sa1
Reviewer #2 (Public review): https://doi.org/10.7554/eLife.100280.3.sa2
Author response https://doi.org/10.7554/eLife.100280.3.sa3

# Additional files

## Supplementary files

MDAR checklist

## Data availability

The trial-by-trial behavioral data that support the findings of this study are publicly available from Zenodo. Matlab scripts for reproducing analysis results and figures are included as part of the shared data package.

The following dataset was generated:

| Author(s) | Year | Dataset title | Dataset URL | Database and Identifier |
|---|---|---|---|---|
| Oor EE, Salinas E, Stanford TR | 2024 | Dataset: Location- and feature-based selection histories make independent, qualitatively distinct contributions to urgent visuomotor performance | https://doi.org/10.5281/zenodo.11391884 | Zenodo, 10.5281/zenodo.11391884 |

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
