## [Editor Report · eLife Assessment]

Oor and colleagues report the potentially independent effects of the spatial and feature-based selection history on visuomotor choices. They outline **compelling** evidence, tracking the dynamic history effects based on their extremely clever experimental design (urgent version of the search task). Their finding is of **fundamental** significance, broadening the framework to identify variables contributing to choice behavior and their neural correlates in future studies.

---

## [Referee Report · Reviewer #1 (Public review)]

Summary:

Oor et al. report the potentially independent effects of the spatial and feature-based selection history on visuomotor choices. They outline compelling evidence, tracking the dynamic history effects based on their clever experimental design (urgent version of the search task). Their finding broadens the framework to identify variables contributing to choice behavior and their neural correlates in future studies.

Strengths:

In their urgent search task, the variable processing time of the visual cue leads to a dichotomy in choice performance-uninformed guesses vs. informed choices. Oor et al. did rigorous analyses to find a stronger influence of the location-based selection history on the uninformed guesses and a stronger influence of the feature-based selection history on the informed choices. It is a fundamental finding that contributes to understanding the drivers of behavioral variance. The results are clear, and the authors convincingly addressed all previously raised concerns, strengthening their conclusions.

---

## [Referee Report · Reviewer #2 (Public review)]

Summary:

This is a clear and systematic study on trial history influences on the performance of monkeys in a target selection paradigm. The primary contribution of the paper is to add a twist in which the target information is revealed after, rather than before, the cue to make a foveating eye movement. This twist results in a kind of countermanding of an earlier "uninformed" saccade plan by a new one occurring right after the visual information is provided. As with countermanding tasks in general, time now plays a key factor in success in this task, and it is time that allows the authors to quantitatively assess the parametric influences of things like previous target location, previous target identity, and previous correctness rate on choice performance. The results are logical and consistent with the prior literature, but the authors also highlight novelties in the interpretation of prior-trial effects that they argue are enabled by the use of their paradigm.

Strengths:

Careful analysis of a multitude of variables influencing behavior

Weaknesses:

Results appear largely confirmatory

Comments on revisions:

The authors have addressed the previous comments.

---

## [Author Response]

The following is the authors’ response to the original reviews

**Public Reviews:**

**Reviewer #1 (Public review):**
Summary:Oor et al. report the potentially independent effects of the spatial and feature-based selection history on visuomotor choices. They outline compelling evidence, tracking the dynamic history effects based on their clever experimental design (urgent version of the search task). Their finding broadens the framework to identify variables contributing to choice behavior and their neural correlates in future studies.Strengths:In their urgent search task, the variable processing time of the visual cue leads to a dichotomy in choice performance - uninformed guesses vs. informed choices. Oor et al. did rigorous analyses to find a stronger influence of the location-based selection history on the uninformed guesses and a stronger influence of the feature-based selection history on the informed choices. It is a fundamental finding that contributes to understanding the drivers of behavioral variance. The results are clear.Weaknesses:(1) In this urgent search task, as the authors stated in line 724, the variability in performance was mainly driven by the amount of time available for processing the visual cue. The authors used processing time (PT) as the proxy for this "time available for processing the visual cue." But PT itself is already a measure of behavioral variance since it is also determined by the subject's reaction time (i.e., PT = Reaction time (RT) - Gap). In that sense, it seems circular to explain the variability in performance using the variability in PT. I understand the Gap time and PT are correlated (hinted by the RT vs. Gap in Figure 1C), but Gap time seems to be more adequate to use as a proxy for the (imposed) time available for processing the visual cue, which drives the behavioral variance. Can the Gap time better explain some of the results? It would be important to describe how the results are different (or the same) if Gap time was used instead of PT and also discuss why the authors would prefer PT over Gap time (if that's the case).

Thanks to Rev 1 for requesting clarification of this important point. As Rev 1 notes, PT is a derived variable, computed for each trial by subtracting the Gap interval from RT (PT=RT‒Gap). While it is true that Gap and PT are correlated (inversely), it is precisely because of the variance in RT that Gap alone is not an adequate (or certainly not the best) predictor of choice outcome. First, note that, if the Gap were fixed, there would still be variance in RT and in outcome, and any dependence of outcome on time would be explained necessarily by the PT. This is true at any Gap. So, clearly, the PT predicts outcome in a way that the Gap cannot. It is easy to see why: the Gap is the part of the RT interval during which no cue information is present, whereas the PT is the part of the same interval during which it is. Therefore, if one accepts the logical premise that the likelihood of a correct choice depends on the amount of time available to view the Cue before making that choice (i.e., the definition of PT), it follows that the relationship between PT and performance should be tighter than that between performance and Gap. And, indeed, this is the case. Mean accuracy declines systematically as a function of Gap, as expected, but its correlation with performance is much weaker than for PT.

Rev 1’s request for a comparison of how accuracy varies as function of PT versus how it varies with Gap has appeared in earlier publications (Stanford et al., 2010; Shankar et al., 2011; Salinas et al., 2014) and we now include it here for the current dataset by adding plots of accuracy versus Gap as a new panel in Fig. 1 (Fig. 1c). That PT (not Gap) better predicts the likelihood of success on a given trial is evident in comparing the tachometric (Fig. 1b) and psychometric curves (Fig. 1c). The tachometric curves vary from chance to asymptotic performance and do so over a short range of PT (~75 ms) with well-defined inflection points identifying key transitions in performance (e.g., from guesses to increasingly informed choices). In contrast, the psychometric function plotting average accuracy versus Gap (Fig. 1c) varies much more gradually, a reduction in temporal definition attributable to the failure to account for the RT’s contribution to determining PT for each trial at a given Gap.

(2) The authors provide a compelling account of how the urgent search task affords(i) more pronounced selection history effects on choice and(ii) dissociating the spatial and feature-based history effects by comparing their different effects on the tachometric curves. However, the authors didn't discuss the limits of their task design enough. It is a contrived task (one of the "laboratory tasks"), but the behavioral variability in this simple task is certainly remarkable. Yet, is there any conclusion we should avoid from this study? For instance, can we generalize the finding in more natural settings and say, the spatial selection history influences the choice under time pressure? I wonder whether the task is simple yet general enough to make such a conclusion.

As Rev. 1 notes, the CO task is a laboratory task that produces large history effects. But importantly, we don't think urgency is causal or essential to the existence of such effects (this is now more explicitly stated in the first section of the Results); it is simply a powerful tool for revealing and characterizing them. As noted in the Discussion, our results are consistent with studies that, based on simpler, non-urgent tasks, demonstrated either reward-driven spatial biases or color priming effects. The CO task uses urgency to generate a psychometric function that time resolves perceptually informed from perceptually uninformed choices, and thereby provides the logical key to disambiguating the simultaneous contributions of perceptual and non-perceptual biases to performance. Such was essential to our demonstration that distinct biases act independently on the same saccade choices.

In a natural setting, we would certainly expect the respective magnitudes of such non-volitional history-based biases to be highly context dependent, but it would be difficult, if not impossible, to discern their relative impact on natural behavior. That said, we think that the biases revealed by the CO task are exemplary of those that would manifest in natural behaviors depending on the real-world context to which such behaviors correspond. Here, it is important to emphasize that the spatial- and feature-based biases we observed were not strategic, on average neither helping nor hindering overall performance. Thus, in the real-world we might expect the expression of similar biases to be an important source of behavioral variance. These observations are now summarized in the penultimate paragraph of the Discussion.

(3) Although the authors aimed to look at both inter- and intra-trial temporal dynamics, I'm not sure if the results reflect the true within-trial dynamics. I expected to learn more about how the spatial selection history bias develops as the Gap period progresses (as the authors mentioned in line 386, the spatial history bias must develop during the Gap interval). Does Figure 3 provide some hints in this within-trial temporal dynamics?

Because it is based on the location of the saccadic choice(s) on previous trial(s), we might expect a signal of spatial bias to be present before and during the Gap period and perhaps even before a trial begins (i.e., intertrial interval). However, because behavioral bias is a probabilistic measure of saccade tendency, we have no way of knowing if such a signal is present during periods devoid of saccadic choices. Note that, for both monkey subjects, average RT exceeded the duration of the longest Gap employed (Fig. 1), and this means that relatively few saccades occurred prior to Cue onset. That said, it's clear in both Figs. 2, 3, and 6 that location bias is evident for saccades initiated at the transition between Gap and Cue intervals (PT=0). Anecdotally, we can report that that spatial bias is evident when we extend our analysis back further into the range of negative PTs (i.e., Gap interval), but the statistics are weak given the paucity of trials at that point. Nevertheless, this is consistent with a bias that exists from the beginning of the trial, as would be expected based on neurophysiological studies from Hikosaka's lab in a simpler but comparable spatial bias task.

Although our data do not unequivocally identify the temporal origin of the spatial bias, they clearly show that the bias is present early (at short PTs) and diminishes rapidly as the perceptual information accrues (at long PTs). Thus, the PT-dependent temporal dynamics that are revealed clearly suggest that spatial and perceptual biases operate over different intra-trial time frames, one decreasing and the other increasing. As mentioned by Rev. 1, Fig. 3 emphasizes this dichotomy.

(4) The monkeys show significant lapse rates (enough error trials for further analyses). Do the choices in the error trials reflect the history bias? For example, if errors are divided in terms of PTs, do the errors with short PT reflect more pronounced spatial history bias (choosing the previously selected location) compared to the errors with long PT?

The short answer is “yes”. Errors generally show a PT-dependent influence of history bias. However, correct and error trials are the result of the same biased dynamics, and analyzing them separately post-hoc does not provide much additional insight about the history effects beyond that provided by the tachometric curves themselves.

To see this, first consider the figure below (Author response image 1). Two tachometric curves conditioned on color history are shown (left). These are the two extreme curves plotted in Fig. 2a, which correspond to the 4S (i.e., 4 repeats of the current target color) and 4D (4 color repeats and then a switch) conditions. Each of these curves already shows the probability of making an error at each PT but, indeed, we can compare the proportions of correct and error trials at short PTs (guesses) and long PTs (informed choices). These are indicated by the bar graphs on the right. Now, the effect of a bias would be to create a difference in success rate between repetitions (4S, blue) and switches (4D, red) relative to the overall, unbiased expectation (indicated by dotted lines). For color-based history, there is no bias at short PT: the proportions of correct choices are almost exactly at the expected chance level (filled bars coincide with dotted line). In contrast, at long PTs, there is a differential effect, but it is due both to a proportion of correct trials that is higher than expected in the 4S case (filled blue bar above dotted line) and to a proportion of correct trials that is lower than expected in the 4D case (filled orange bar below dotted line). This is exactly as one would expect if the current choice was biased by target color history.

A similar analysis can be done for location history (Author response image 2, which shows the two extreme curves from Fig. 2e). In this case the bias is much stronger at short PTs, and the difference between repeats (4S, blue) and switches (4D, red) is largely explained by a proportion of correct choices that is much higher than expected by chance in the 4S condition (filled blue bar well above dotted line). This makes sense, because a rewarded location is likely to become the next guess, so if the target happens to appear again at that same location, the subsequent guess is more likely than chance to be correct. At longer PTs, the differential effect is smaller, as would be expected for more informed choices, but it is again driven by the 4S condition. Importantly, in the case of location the total number of S trials is much smaller than the total number of D trials (because a target-location repetition has a probability of 0.25 only), so it only makes sense to compare the proportions of correct (or error) trials, not the absolute numbers, between those conditions.

**Author response image 2. sa3fig2:** 

In summary, although it is possible to examine the separate dependencies of correct and error trials on history and PT, the distinction is not very useful. Only the frequency of errors relative to that of correct choices makes complete sense, not so much, say, the frequency of short PT errors relative to that of long PT errors.

**Reviewer #2 (Public review):**
Summary:This is a clear and systematic study of trial history influences on the performance of monkeys in a target selection paradigm. The primary contribution of the paper is to add a twist in which the target information is revealed after, rather than before, the cue to make a foveating eye movement. This twist results in a kind of countermanding of an earlier "uninformed" saccade plan by a new one occurring right after the visual information is provided. As with countermanding tasks in general, time now plays a key factor in the success of this task, and it is time that allows the authors to quantitatively assess the parametric influences of things like previous target location, previous target identity, and previous correctness rate on choice performance. The results are logical and consistent with the prior literature, but the authors also highlight novelties in the interpretation of prior-trial effects that they argue are enabled by the use of their paradigm.Strengths:Careful analysis of a multitude of variables influencing behaviorWeaknesses:Results appear largely confirmatory.
**Recommendations for the authors:**

**Reviewer #1 (Recommendations for the authors):**
(1) The authors provide comprehensive accounts of the urgent search task in multiple places in the manuscript. But the description can be simpler and more consistent throughout. I found it confusing when the authors compared their task with previous search tasks used by Bichot and Schall, McPeek et al. I believe the authors wanted to explain that it is not just the urgency but the fact that the target color being randomly interleaved also contributes to the pronounced history bias in their task. I appreciate their thorough comparison with previous studies but it can be distracting or lose focus. It might read better if this statement can be expanded in the Discussion, not in the Results (lines 366-376).

We thank the reviewer for pointing this out. We agree that the paragraph in question was ambiguous and appeared to elaborate a Discussion point, which was not our intent. Indeed, as the reviewer noted, the main point was that the randomization of the target colors (and not urgency) is the critical aspect of the task that makes it surprisingly difficult for the monkeys. We have revised the paragraph to emphasize this conclusion and the two empirical results from our own data that support it. The agreement with prior studies, which is somewhat tangential, is now briefly mentioned at the end of the paragraph. It should now be clear that the text mainly describes current data that are relevant to the interpretation of the main results.

(2) It's important to state that feature-based selection history bias is not merely due to the monkey's intrinsic bias to one color over the other (red vs green). The authors did a nice job controlling that, as mentioned in Methods (lines 194-196) and supplementary figure (Figure 1 - Figure Supplement 2). It would be helpful for readers to read in Results as well.

Thank you for the suggestion. We now mention this in the second section of the Results.

(3) D trial examples for the location history in Results can be confusing to readers (lines 407-409; left-left-right, up-up-left). The examples in Methods (lines 224-229; left-up-right, up-down-left) are better to convey the preceding (different) trials can be of any kind.

Indeed. Both types of example are now mentioned in the Results.

**Reviewer #2 (Recommendations for the authors):**
I have only minor comments:(1) In the abstract, I'm not sure what "when combined" means in the last sentence. What is combined? Selection history and stimulus salience? If so, this is not very clear. Also, it might be nice to end the abstract on how the study addresses the three components of attention that the abstract started with in the first place (salience, task, and history). Otherwise, I spent multiple abstract reads (before even reading the rest of the paper) trying to see whether indeed the paper addresses the three components of attention that were so prominently described at the beginning of the abstract or not. And, I still could not convince myself of whether all three were addressed by the study or not (I then resorted to proceeding with a reading of the rest of the paper).

Thanks for pointing this out. We have reworded the abstract to clarify that we are focusing on selection history, not salience or top-down attention.

(2) Line 72: isn't stimulus location still a feature????

Our nomenclature here is intended to be consistent with the commonly applied distinction between “spatial” and “feature” -based attention that underscores the distinct mechanistic underpinnings of “where” and “what”.

(3) Lines 76-79: I'm very confused here. The part about "guesses can be strongly biased toward an arbitrary location early on". However, I expected the later part of the sentence to still stick to location and mention what the temporal dynamic is. Instead, it discusses perceptual bias, which I presume is the color thing. So, the net result is that I'm a bit confused about how *both* location and color behave in *both* early and late times.

We have rewritten the end of this paragraph to clarify when and how location and feature biases manifest in behavior. It may be useful to note the following. The tachometric curve describes different types of choices distinguished by their timing, guesses at short PTs vs informed decisions at long PTs. However, this also corresponds to the degree to which perceptual information becomes available over time *within a single trial*. Namely, perceptual information is initially absent but arrives later on. The revised text now reflects this distinction, making the logic for the expected results clearer.

(4) Last paragraph of the introduction (lines 80-82): it would be helpful to justify here why the psychophysics were done in monkeys in this study, instead of humans.

We now allude to the reason these studies were done in monkeys but feel that more elaboration of this point is better left to Discussion. The Discussion now more explicitly states that the current data are closely related to neurophysiological studies of spatial attention and color priming in monkeys (beginning of 4th paragraph).

- Line 389: this kind of formulation is much clearer to me than lines 76-79 mentioned above.

As noted, the above-mentioned section has been revised.

- I'm a bit confused by Figure 4 in the sense that some of the effect sizes are not too different from Figure 2, even when there are some intermediate inconsistent trials. I guess the problem is aggravated by the different axis ranges in Figures 2, and 4.

All the 1S and 1D data points are the same in both figures, as they should, but the problem is that, otherwise, the two figures are just not comparable. Apples and oranges. To see this, note that the trends for the difference between S and D conditions should go in opposite directions as trials go further into the past, and indeed they do. In Figures 2c, f, the differences between 1S and 1D results are small, and those between 4S and 4D results are the largest because both S and D effects grow away from the average with more repetitions. In contrast, in Figure 4b-d, the differences between S and D shrink as the effect of a single trial becomes more distant (differences are largest between 1S and 1D results, smallest between 1S9x and 1D9x results). The only slightly ambiguous trend is that of Figure 2g, because the S data are more noisy. We have expanded the text surrounding Figure 4 to highlight the different expected trends for this analysis in contrast to that presented in Figure 2. This should clarify the qualitative difference between the two.

- On a related note, it is odd that the summary figures (e.g., Figures. 2, 4, etc) are vertically aligned such that the dependent measure is on the x-axis rather than the y-axis. For example, looking at Figure 2, it would make much more sense if panels b-d and f-h were rotated by 90 deg, such that the vertical axis is indeed the low asymptote or high asymptote or RT. This would directly correlate with the same data in panels a and e in the same figure and would be much easier to follow. Then, later in the paper, Fig. 8 suddenly does the dependent measure on the y-axis, as I said. I think it can help to use similarly consistent plotting approaches across all (or most) analyses.

We tried other formats but settled on the current one because we felt it made it (slightly) easier to compare the patterns across history conditions between any two of the 6 bar graphs in each figure (in Figs 2, 5, 6), in part because it prevents any confusion with the PT axes. As this does not make a substantial difference either way, we prefer to maintain the present arrangement. Additional labels are now included, which should make the figures a bit more friendly.

- At the beginning of the paper, I was under the impression that this will really be a free viewing search task (e.g., Wolfe search arrays or old Nakayama search arrays), but then it became clear later that it was still an instructed task, with the only difference being that the target onset is now 4 targets. I think this distinction should be clarified very early on, in order to avoid confusion by the readers. The reason I say this is that with enforced fixation, there are other factors in this task that come into play, like the monkey's individual microsaccade rates etc, which can modulate performance since they also have a form of countermanding that is like the one imposed by the compelled saccade task. So, better alert the readers to the context of the task early on.

Thanks. We have provided additional detail when introducing the task for the first time in the Introduction, along with a citation to an earlier publication in which the specific task is described. There should be no ambiguity now.

**Reviewing Editor Comments:**
Short Assessment:This important study makes compelling use of the monkey animal model to capture the long-time course over which trial history affects decision-making under time pressure, showing decisions are affected by the stimulus sequence extending back as many as four trials previously.Summary:Decision-making is variable, but how much of this variability can be accounted for by the immediate previous history is not well known. Using an "urgent" saccade, Oor et al manipulated how much time monkeys had to process evidence, and evaluated what they did when there was too little time to make an evidence-based decision. They report that the history affected performance as far back as 4 previous trials and that different aspects of the stimulus history (color and location) affected performance differently.Strengths:The key strengths of this paper are that the monkey paradigm permitted a study under highly controlled conditions with stable performance across sessions and enough trials to conduct the history analysis farther back in time than is possible with smaller data sets. While the fact that prior history affects decisions was previously known, this study provides a careful quantification of the effect -- which proves to be quite large - as well as an assessment of both location and feature histories in combination with each other. The manuscript is well-written and easy to follow.Weaknesses and recommendations for the authors:(1) The figures are lovely but could use some more text/design elements to clarify, and there is space to do so. e.g., in Figure 2, there could be titles to indicate that the top row involves the color history and the bottom row involves location history. The information is there, in the y labels of panels B and F, but it takes a while to see that.

Done. Titles have been added to Figure 2 and several others.

(2) Furthermore, the abbreviations 1D, 4S, etc are explained in the legend but it seems there is room to spell them out or include a graphic to indicate what they mean.

The labels 1D, 4S, etc are difficult to spell out because each one represents multiple conditions; for instance, 2S may correspond to green-green or red-red target colors, and so on. Figure legends have been edited to more clearly indicate that S and D labels correspond to repeat and switch trials, respectively, and that the associated number indicates how far back the history goes.

(3) The terms "low asymptote" and "high asymptote" could be indicated in a graphic of a tachymetric function, smoothing the transition to the rightmost panels. (Consider also alternative terms - perhaps "floor" and "ceiling" might be more readily understandable than asymptote to the student reader??).

Thanks for the suggested terms, “floor” and “ceiling”, which we’ve adopted. They are indeed more natural. Figure 2a now indicates that floor and ceiling accuracies correspond to opposite ends of the PT axis.

(4) The units for the asymptotes are not indicated - I assume these are "% correct" but that would be helpful to clarify.

Yes. Units for floor and ceiling (and RT) are now indicated in all figures.

(5) Figure 3 - "PT", and "1S-1D" could be spelled out, and the meaning of the two colored traces could be in the figure itself rather than only in the legend. Similar suggestions apply about labeling, abbreviations apply in subsequent figures.

PT is now spelled out in all figures other than Figure 1, and labels for the two traces were added to Figure 3. Thanks for all the detailed suggestions.